# Glioblastoma disrupts cortical network activity at multiple spatial and temporal scales

Jochen Meyer [1,2,3] ✉, Kwanha Yu[3,4,5], Estefania Luna-Figueroa[4], Benjamin Deneen [3,4,5] & Jeffrey Noebels [1,2,3,6] ✉

The emergence of glioblastoma in cortical tissue initiates early and persistent neural hyperexcitability with signs ranging from mild cognitive impairment to convulsive seizures. The influence of peritumoral synaptic density, expansion dynamics, and spatial contours of excess glutamate upon higher order neuronal network modularity is unknown. We combined cellular and widefield imaging of calcium and glutamate fluorescent reporters in two glioblastoma mouse models with distinct synaptic microenvironments and infiltration profiles. Functional metrics of neural ensembles are dysregulated during tumor invasion depending on the stage of malignant progression and tumor cell proximity. Neural activity is differentially modulated during periods of accelerated and inhibited tumor expansion. Abnormal glutamate accumulation precedes and outpaces the spatial extent of baseline neuronal calcium signaling, indicating these processes are uncoupled in tumor cortex. Distinctive excitability homeostasis patterns and functional connectivity of local and remote neuronal populations support the promise of precision genetic diagnosis and management of this devastating brain disease.

Glioblastoma (GBM), the most aggressive form of brain cancer, exploits the complex interplay between neurons and glia to create a microenvironment favorable for its own expansion yet hostile to network excitability homeostasis, impeding efforts to achieve long-term clinical remission. Nearly 60% of GBM cases present with seizures and develop pharmacoresistant tumor-related epilepsy (TRE), substantially reducing the quality of life despite tumor resection[1-3]. At the leading edge, a complex sequence of molecular, cellular, and synaptic remodeling in the tumor microenvironment (TME)[4-6] with early loss and impairment of interneurons[7-9] promotes epileptogenesis and impairs cognitive processes. Distinguishing between healthy and epileptic tissue during tumor resection, as well as defining whether network hyperexcitability in areas more distant than 1 mm from the tumor margin is recruited and how this

remodeling can be suppressed to regain healthy cortical function remain major treatment challenges.

Recent work has demonstrated that GBM progression can unfold nonlinearly over time[10] and is intimately interconnected with TRE in a multifactorial fashion[6,11-15]. GBM preferentially kills or dysregulates inhibitory interneurons[16] and modifies synaptogenesis[4,6,8,17,18] in a tumor driver gene-dependent manner[19]. We previously showed in an immunocompetent murine model that GBM causes a gradual increase of electrographic cortical hyperexcitability coinciding with a reproducible pattern of peritumoral neuronal cell death, particularly in fast-spiking parvalbumin-positive (PV+) interneurons, degradation of perineuronal nets with widespread microglial activation and waves of spreading cortical depolarization that precede the onset of seizures, all providing evidence of this excitotoxicity[8]. Similar pathology and

[1]Department of Neurology, Baylor College of Medicine, Houston, TX, USA. [2]Department of Neuroscience, Baylor College of Medicine, Houston, TX, USA. [3]Center for Cancer Neuroscience, Baylor College of Medicine, Houston, TX, USA. [4]Center for Cell and Gene Therapy, Baylor College of Medicine, Houston, TX, USA. [5]Department of Neurosurgery, Baylor College of Medicine, Houston, TX, USA. [6]Department of Human and Molecular Genetics, Baylor College of Medicine, Houston, TX, USA. ✉e-mail: jfmeyer@bcm.edu; jnoebels@bcm.edu

GBM-induced changes in neurovascular coupling and neural activity dysregulation using human glioma cells implanted in mouse cortex offer additional evidence that improved understanding of malignant progression in diverse tumor subtypes will enable novel therapeutic opportunities[9,11,12,20]. With the advent of precision genetic profiling of human GBM[21], correlating the heterogeneity and pace of ambient pathophysiology with the tumor's molecular profile may translate into improved individualized management of GBM cortical comorbidity.

GBM corrupts the synaptic microenvironment to facilitate its spread, and peritumoral glutamate is a well-established candidate mechanism underlying GBM hyperexcitability[22–24]. Excess glutamate release is an expected feature of a synaptic network with an elevated E/I ratio; however, the precise onset, temporal progression, and spatial extent of neuron-tumor crosstalk, i.e., feedback interactions between tumor cells and neurons, relative to local and distant synaptic activity, and its variation among different tumor types, has never been determined. To address this variability, we compared a previously reported in utero electroporation (IUE) model of glioma (henceforth referred to as 3xCR)[4,8,19] with tumors that included IUE addition of GPC6, a glypican family member of glial-secreted factors that promote synaptogenesis[25,26]. GPC6 was specifically linked with the formation of excitatory synapses. Moreover, recent studies demonstrate that other glypican family members can promote glioma growth, reduced survival, and TRE[19,27]. We combined chronic cortical imaging of peritumoral neural activity using calcium and glutamate reporters over a prolonged 3-month period of tumor expansion in awake mice. Simultaneously recorded EEG, locomotion, whisker, and eye movements allowed us to control the behavioral and attentional state of the animals during imaging. We analyzed peritumoral activity both mesoscopically at low spatial magnification/high temporal resolution covering both hemispheres, and at high magnification (two-photon excitation microscopy) in the same animals to pinpoint the distance of individual neurons from tumor cells at the leading edge. For cellular-resolution imaging, we focused on glutamatergic neurons, because, in our model, there was disproportionally more cell death in some GABAergic cell types near the tumor edge[8]. Our data reveal a nonlinear sequence of changes in functional network connectivity over time, a clear genetic dissociation between the speed of tumor invasion, and a strong correlation between accelerated local cortical tumor expansion and neuronal activity gradients along distance from the tumor margin. Spatially and temporally aberrant glutamate accumulation surpassed and outpaced the dynamics of neuronal calcium levels, providing pioneering evidence that these processes are not tightly coupled.

## Results

### Tumors with and without GPC6 generate distinct cortical infiltration dynamics

Our previous studies demonstrated how tumor intrinsic factors influence the local neighboring neurons and the larger synaptic network at critical stages of tumor hyperexcitability[4,8,19,28,29]. In an effort to gain more granular insights into these dynamics, we developed a multimetric, longitudinal imaging assay system for prolonged, regular monitoring of tumorigenesis, as shown in Fig. 1.

To assess the potential contributions of GPC6 in gliomagenesis, we first mined its expression in human GBM. Utilizing established, publicly available datasets, we observed that GPC6 expression is significantly increased in both low-grade glioma (LGG) and GBM compared to normal brain tissue (Fig. 2A). To evaluate whether GPC6 could be functionally relevant to gliomagenesis, we compared the survival of human patients with differential GPC6 expression, finding that high GPC6 expressing patients demonstrated significantly worse survival (Fig. 2B). Lastly, to validate the presence of GPC6 in human patients, we tested for GPC6 protein in a glioma tissue microarray, finding enhanced expression in tumor samples compared to normal brain

(Fig. 2C). Together, these findings suggest that GPC6 could be functionally relevant toward promoting gliomagenesis.

To test for this, we overexpressed GPC6 in our IUE mouse glioma model (Fig. 2D). Of note, in our model GPC6 expression was introduced into all tumor cells, whereas in the human samples GPC6 was likely not equally as homogeneously expressed throughout the tumor. We observed that adding GPC6 significantly accelerated tumor-associated death with a reduction in median survival of 2 weeks (median survival 3xCR vs GPC6: 97 vs 83 days, respectively). Interestingly, this mimics the patterns observed in the human data (Fig. 2A). In concordance with the quickening of tumor burden, we observed a significant increase in BrdU incorporation (Fig. 2E), suggestive of early accelerated tumor growth.

To gain a molecular perspective on the changes GPC6 elicits on a tumor brain, we performed bulk tissue RNA-Sequencing (RNA-Seq) from endpoint 3xCR and GPC6 tumors and identified 1917 differentially expressed genes ($p$ value <0.05, Fig. 2F). Of these, 1350 were upregulated in GPC6 tumor brains. Gene ontology analysis (Supplementary Fig. 3) across different ontology databases revealed that the top categories were associated with synapses, neurons, or seizure/hyperexcitability. Additional analyses of publicly available datasets confirmed GPC6 expression in tumor cells only (Supplementary Fig. 4), as well as reduced survival depending on GPC6 expression across different glioma subtypes (Supplementary Fig. 5) To assess changes in the neighboring synaptic microenvironment, we stained peritumoral sections for excitatory and inhibitory synapses in developing tumors at P30 (Fig. 2G, H) and found a significant increase in excitatory synapses but no significant change in inhibitory synapses in the presence of GPC6. The aforementioned analyses were performed at P30 tumor brain samples revealing the early synaptic imbalance mediated by GPC6 overexpression, however the molecular mechanism of GPC6 overexpression tumor effects remain to be disentangled from complex homeostatic network compensatory mechanisms at later disease stages.

Next, we analyzed the spatiotemporal patterns of cortical tumor invasion, comparing the 3xCR tumor model with and without the overexpression of GPC6. As a measure of tumor infiltration or motion dynamics, we calculated the CV/day (i.e., coefficient of variation of the pixel-wise daily rate of intensity fluctuation) between serial tumor fluorescence samples, a challenging assay due to potential changes in transmission through the cranial imaging window (see methods, Fig. 3A, B; see also Supplementary Fig. 6 demonstrating consistent clarity of the cranial window preparation for this analysis over many weeks). In contrast with the early large dynamic changes in tumor fluorescence in the 3xCR model, the representative GPC6 animal (Fig. 3B) exhibited significantly attenuated variations in tumor infiltration over time (example images calculated from four time points between P62 and 83). We analyzed a total of 11 3xCR and 8 GPC6 animals and found that the CV/day was, on average, 52% lower for GPC6 than for 3xCR tumors (0.031 ± 0.005 sem vs 0.065 ± 0.011 sem, $p = 0.033$, WR test, Fig. 3C). In addition, quantification of local tumor coverage rates (in $\mu m^2$/day) as a function of time revealed early elevation of GPC6 tumor expansion, and sustained expansion rates of 3xCR tumors, whereas they were lower at late stages in GPC6 animals (Fig. 3D). In each recording session, we verified the presence of tumor cells and the outline of the tumor margin by acquiring cellular-resolution two-photon z-stacks (see examples in Fig. 3E). These results indicate that 3xCR tumors attained more rapid infiltration rates locally at the cortical surface than GPC6 tumors. GPC6 tumors appeared to expand aggressively at first, but less consistently later, while 3xCR tumor continued to expand at the cortical surface. This intriguing and unexpected non-uniform difference in expansion rates would not have been detected without chronic in vivo imaging and warrants further investigation in future studies.

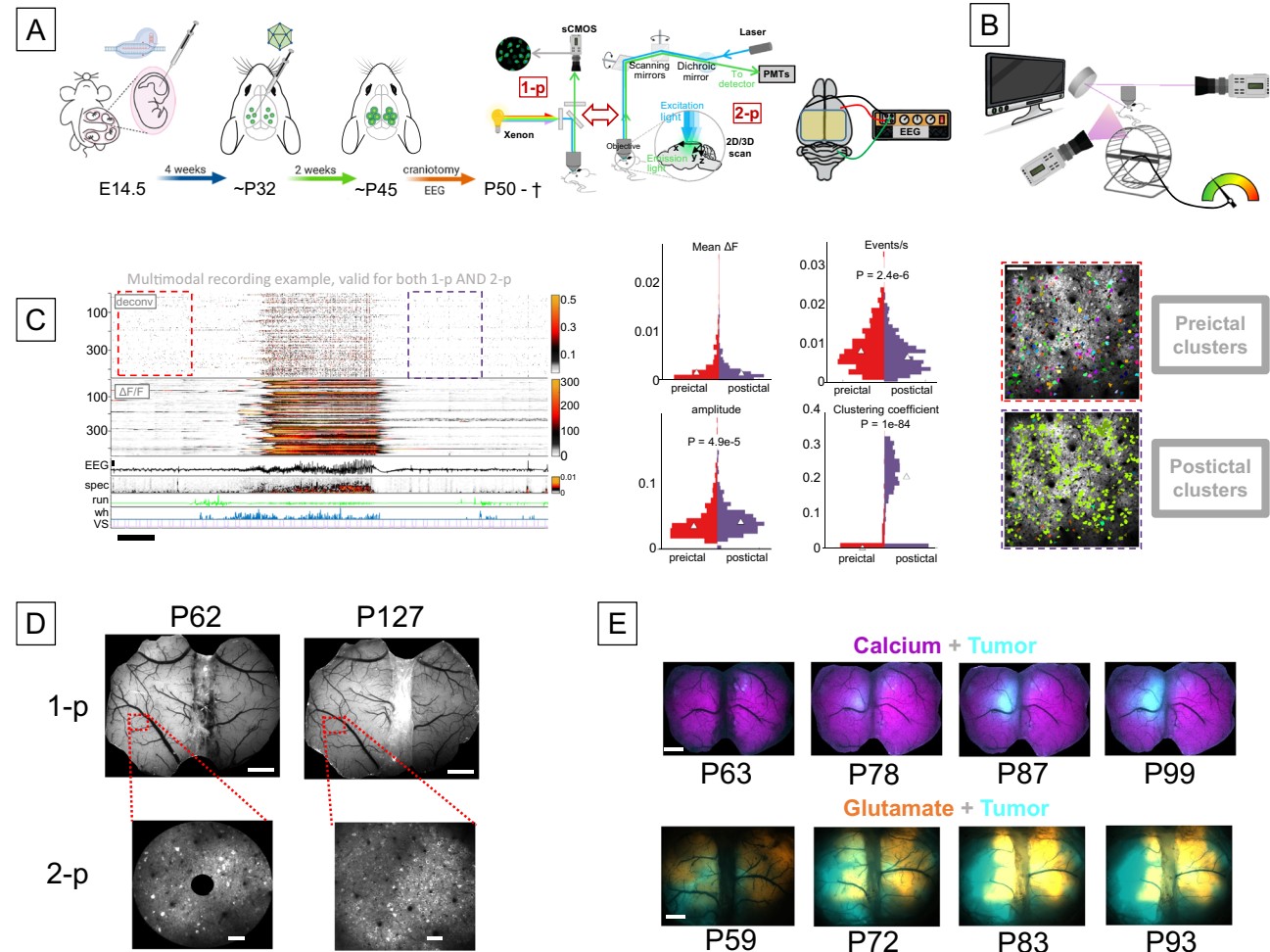

**Fig. 1 | Integration of an IUE murine GBM model with multimodal recordings of cellular, widefield fluorescence, and behavioral parameters. A** Schematic illustration of experimental design. IUEs are performed on E14.5 mouse embryos. At 1 month of age, AAV viruses are injected intracortically. After 2 weeks, a bilateral ~7 × 5 mm cranial window and two-channel EEG electrodes were installed on the mouse's cranium. **B** Schematic illustration of multimodal monitoring of the animal's behavioral state: Running wheel velocity, pupil and eye movements, and whisking/frontal body movements were recorded. An LCD monitor was used to display visual stimuli. **C** Time series raster plot of representative two-photon imaging recording showing baseline (red dotted box), seizure, and postictal activity (black dotted box). Traces show ΔF/F calcium activity, EEG voltage, and mouse behavior illustrated in (**B**). Scale = 10 s. Violin plots display distributions of

deconvolved activity metrics extracted from 150 s of the baseline (red dashed rectangle), and 150 s of postictal (purple dashed rectangle) periods. White triangles = median. Statistical significance between pre- and postictal periods was computed using the Wilcoxon rank-sum test, $n$ = 432 neurons. Not significant for mean DF. Preictal and postictal clusters shown to the right were computed from the same recordings. Neurons belonging to the same cluster share the same color. Scale = 0.1 mm. **D** Long-term optical stability of cranial windows: widefield (top) and 2p (bottom) images from the same animal at P62 and P127. Scale (top) = 1 mm, scale (bottom) = 0.1 mm. **E** Examples of dual-indicator widefield images of different tumor fluorescence and activity indicators over time. Top row: thy1-GCaMP6s line and tumor pseudocolored in magenta and cyan, respectively. Bottom row: iGluSnfr and tumor pseudocolored in yellow and cyan, respectively. Scale = 1 mm.

## Cortical baseline excitability and glutamate dynamics are uncoupled during GBM progression

Given the specific increase in excitatory glutamatergic synapses in GPC6 tumors at P30 (Fig. 2), we sought to investigate whether the dynamics of glutamate release and calcium mobilization (representing baseline somatic and synaptic electrical activity) would differ at time periods between P45–P70, when AAV-mediated glutamate and calcium indicators were sufficiently expressed, versus after P70, when mortality increases, and how this would differentiate network activity between 3xCR tumors and those additionally expressing GPC6. The analysis presented here (Fig. 4) does not directly correlate calcium and glutamate spatially with tumor position or size, rather it focuses exclusively on calcium and glutamate changes across disease progression. To this end, we virally introduced genetically encoded fluorescence indicators for calcium (or alternatively used thy1-GCaMP6s expressing mouse line, Jackson #024275) and glutamate

(AAV-FLEX-iGlusnfr + AAV-Ef1α-Cre or AAV-CamKIIα-Cre, see methods and Supplementary Table 1). We compared snapshots of calcium and glutamate baseline fluorescence during quiet wakefulness, without extracting fast activity metrics like we do for calcium signals in the following sections, because glutamate and calcium reporter kinetics vary, and interpreting this comparison would have been challenging. In 3xCR animals, we observed a rapid and spatially heterogeneous elevation of calcium activity taking place before P70, followed by an indolent period with little further change (<10% between time points) in baseline calcium (see Fig. 4A for images from a representative 3xCR animal, processed via the same approach shown in Fig. 3). In contrast to the neuronal calcium signals, we found a steadily changing glutamate signal between similar time points (Fig. 4B). Glutamate began to accumulate slowly locally, then intensified strongly after P70. There was a striking continued dynamic instability (regions of dark red and blue in colored images corresponding to large CV fluctuations) of

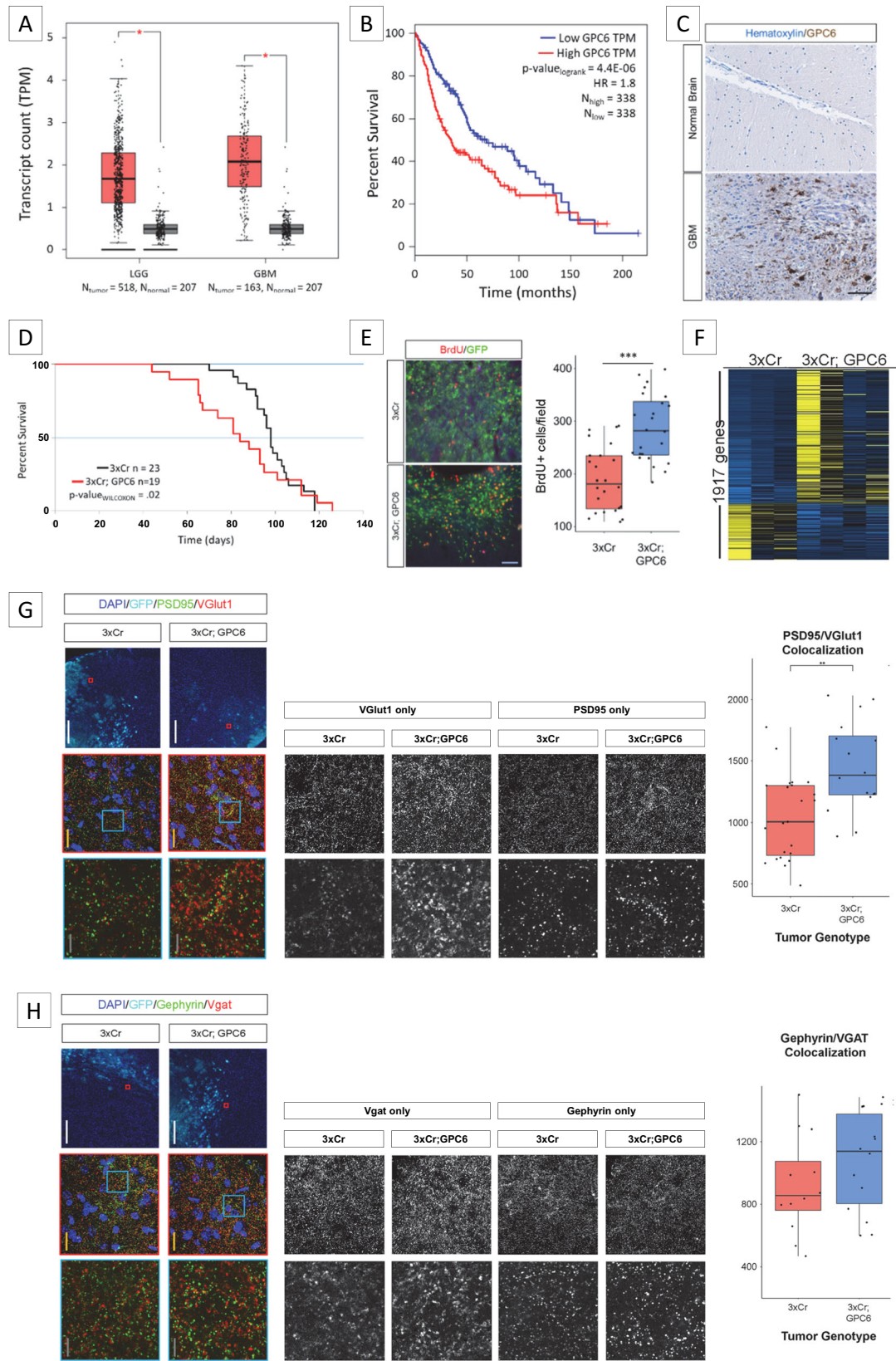

glutamate levels throughout the disease progression. Throughout the duration of the imaging study, changes in calcium baseline fluorescence appeared consistently smaller than the changes in glutamate-reporter brightness in 3xCR tumor animals (Fig. 4D).

In GPC6 tumor brains, there was a stronger modulation in baseline calcium signal before P70 (Fig. 4C), compared to 3xCR. GPC6 calcium accumulated at earlier time points (<P70, Fig. 4E, calcium, blue vs. red), whereas it did not change over time around 3xCR tumors (Fig. 4E, calcium, 3xCR, <P70 vs >P70).

This result demonstrates a dynamic change in extracellular glutamate levels during GBM progression that correlates poorly with neuronal calcium baseline levels. As shown in Fig. 1E (bottom row) and

**Fig. 2 | GPC6 is enriched in human and murine GBM. A** GPC6 expression based on transcript count per million (TPM, log scale) from human datasets comparing low-grade glioma (LGG), glioblastoma (GBM), to normal non-tumor brain. Source data from Gepia (PMID: 28407145) using TCGA normal and GTEx data for LGG and GBM datasets. Asterisk signifies *p* values <0.01 (one-way ANOVA). **B** Kaplan–Meier survival analysis of human patient cohorts of LGG and GBM patients. Low and high TPM cutoff were set at below and above the 50th percentile, respectively. HR hazard ratio. **C** Immunohistological staining for GPC6 on human GBM tissue (below) along with normal brain control (above). Scale bar = 100 μm. Representative image of six different human GBM samples. **D** Kaplan–Meier survival analysis comparing 3xCR (black) and GPC6 (red) tumor-bearing mice. *p* value calculated through Wilcoxon signed-rank test. **E** Representative immunofluorescence of BrdU incorporation (red) in tumor sections (green). Quantification of BrdU+ cells per field (100,000 μm²) *p* values calculated by one-tail student *t*-test, *p* = 3.7e-6. *** <0.001. *N*(biological) = 4 brains. *N*(technical) = 6 images per brain. **F.** Heatmap of differentially expressed genes between 3xCR and GPC6 tumors brains. Each column represents a single brain. The first three columns are 3xCR tumors. The last four columns are GPC6 tumors. Yellow−high expression. Blue−low expression. **G** Representative images from immunohistofluorescence images of PSD95 (green) and vGlut1 (red) staining around tumor margins (cyan). Quantifications of colocalization of Psd95 and vGlut1. *p* value calculated by one-tail student *t*-test, *p* = 0.0045. ** <0.01. White scale bar = 300 μm; yellow scale bar = 20 μm; gray scale bar = 5 μm. *N*(biological) ≥ 3 brains. *N*(technical) = 6 images per brain. **H** Representative images from immunohistofluorescence images of Gephyrin (green) and Vgat (red) staining around tumor margins (cyan). Quantifications of colocalization of Gephyrin and Vgat. *p* value calculated by one-tail student *t*-test. No significant difference. White scale bar = 300 μm; yellow scale bar = 20 μm; gray scale bar = 5 μm. *N*(biological) = 3 brains. *N*(technical) = 4 images per brain. Source data are provided as a Source Data file for **E**, **F**, **G**, **H**. All box-whisker plots (**A**, **E**, **G**, **H**) center on the median; the bounds of the boxes mark the upper and lower quartile; the whiskers extend to the upper and lower extremes (1.5x interquartile range from the upper and lower quartiles).

Supplementary Fig. 7, these distant nonlinear glutamate accumulations may extend several millimeters away from the tumor margin, even including large areas remote from the tumor in the contralateral hemisphere.

### Mesoscale calcium activity patterns are significantly disturbed depending on the local tumor expansion rate

Specific types of GBM can display nonlinear proliferation patterns over time[10], and this might influence their effects on surrounding neuronal physiology in their microenvironment. We asked whether spontaneous peritumoral aggregate neural activity within <7.5 mm of the tumor edge was significantly modulated with the advent of tumor cell cortical invasion. Specifically, we analyzed whether changes in ongoing activity levels and patterns depend on concurrent tumor expansion rates. To investigate this, we bifurcated the data into two bins of fast and slow-growing tumors at above or below $10^5$ μm²/day, respectively (Fig. 3D). When we quantified all the measured events in a given frame during fast and slow spreading stages in a representative 3xCR and GPC6 brains (Fig. 5A), we observed significant differences in activity patterns.

To gain a more granular view of activity level and mesoscopic neural population pattern differences between fast and slow tumor expansion phases, we extracted and analyzed individual calcium events corresponding to simultaneous electrical activity of local neuronal ensembles[30–32]. We identified differences in activity patterns between 3xCR and GPC6 animals that depended on whether concurrent local tumor expansion was slow or fast (Fig. 5B). The mean activity (ΔF/min) was 2.3-fold greater in 3xCR than in GPC6 brains during slow, but not fast, expansion (*p* = 0.008 and *p* = 0.15, respectively, WR test), indicating an unexpected shortage of overall synaptic and somatic activity in GPC6 mice during slow tumor expansion. The amplitudes of isolated 3xCR calcium events were elevated in both slow and fast tumor expansion states compared to GPC6 (+102 and +165%, respectively, *p* = 0.008 for both comparisons, WR test). Mean event rates were 27% lower in 3xCR animals than in GPC6 animals when tumors expanded rapidly, but not slowly (*p* = 0.008 and *p* = 0.095, respectively, WR test), whereas events lasted 176% longer in 3xCR during slow expansion but not fast (*p* = 0.008 and *p* = 0.056, respectively, WR test). Thus, it appeared that when the tumor expansion rate was high, overexpression of GPC6 altered the whole-FOV activity patterns qualitatively by favoring more frequent but smaller events, whereas when tumors expanded slowly, tumors with GPC6 had reduced overall calcium activity.

### Neuronal excitability depends on tumoral GPC6 expression, distance, and expansion rate

Having established a differential influence of GPC6 on whole-FOV neuronal activity changes as a function of local cortical tumor expansion rate, we next asked whether ongoing calcium activity patterns in neuronal populations also depended on distance from tumor cells. We calculated distance bands in 0.75 mm increments, as shown in Fig. 5C for 3xCR and GPC6, respectively, averaging the ΔF/F signal concentrically from the tumor border (outlined in white lines). We determined significant differences in activity metrics near versus far from the tumor edge by dividing the average pixel values inside the distance band closest to the tumor (<0.75 mm) by the average pixel values inside the distance bands located further than 3 mm from the edge (see Fig. 5C example scatter plots for all four metrics from one 3xCR and one GPC6 animal. Note the relative lack of a clear relationship between tumor expansion rate and near vs far metric ratios in the GPC6 animal). Based on the concurrent expansion rate, these ratios were grouped into "slow" and "fast" and compared to each other and across genotypes (pooled data from five 3xCR and five GPC6 animals). The mean ΔF/F ratio (near vs far) in 3xCR animals during periods of fast tumor expansion was, on average, 2.84 ± 0.34 sem, versus 1.55 ± 0.19 sem during slow expansion (KW/mc test, *p* = 0.013, Fig. 5D, left). Whereas these results from 3xCR tumor animals indicate that overall calcium events were progressively diminished at increasing distances from the tumor, in GPC6 animals there was no such difference in near/far ratios between slow and fast expansion periods. Similar results were obtained for the near/far ratios of event duration between fast and slow 3xCR tumor expansion periods (mean 1.71 ± 0.15 sem vs 1.21 ± 0.06 sem, *p* = 0.022, KW/mc test) and for amplitude (2.13 ± 0.11 sem vs. 1.52 ± 0.127 sem, *p* = 0.027, KW/mc test). Mean ΔF/F, duration, and amplitude ratios were also higher in 3xCR than in GPC6 animals during fast expansion (*p* = 0.03; *p* = 0.0024; *p* = 0.003, KW/mc test, respectively). Event rate ratios were not significantly different across tumor expansion rates or genotypes.

These results demonstrate that mesoscopic peritumoral neural activity patterns can be differentially modulated according to distance from the tumor margin at a sub-millimeter scale, depending on the contemporaneous proliferative state and specific genetic makeup of the tumor. Mean calcium activity, event amplitude, and duration were generally stronger close to 3xCR tumors under rapid spread conditions, than further away from the tumor edge. GPC6 tumors did not seem to influence calcium event strength in this distance-dependent manner, and under fast tumor expansion rates, all calcium metrics−except event rate, which did not depend on distance−were significantly less correlated with tumor distance than in 3xCR animals.

### GPC6-dependent changes in cellular level activity patterns inside versus outside the tumor margin

One-photon widefield imaging enabled the monitoring of large cortical areas simultaneously, which proved useful for measuring activity changes around tumors with a circumference of several mm.

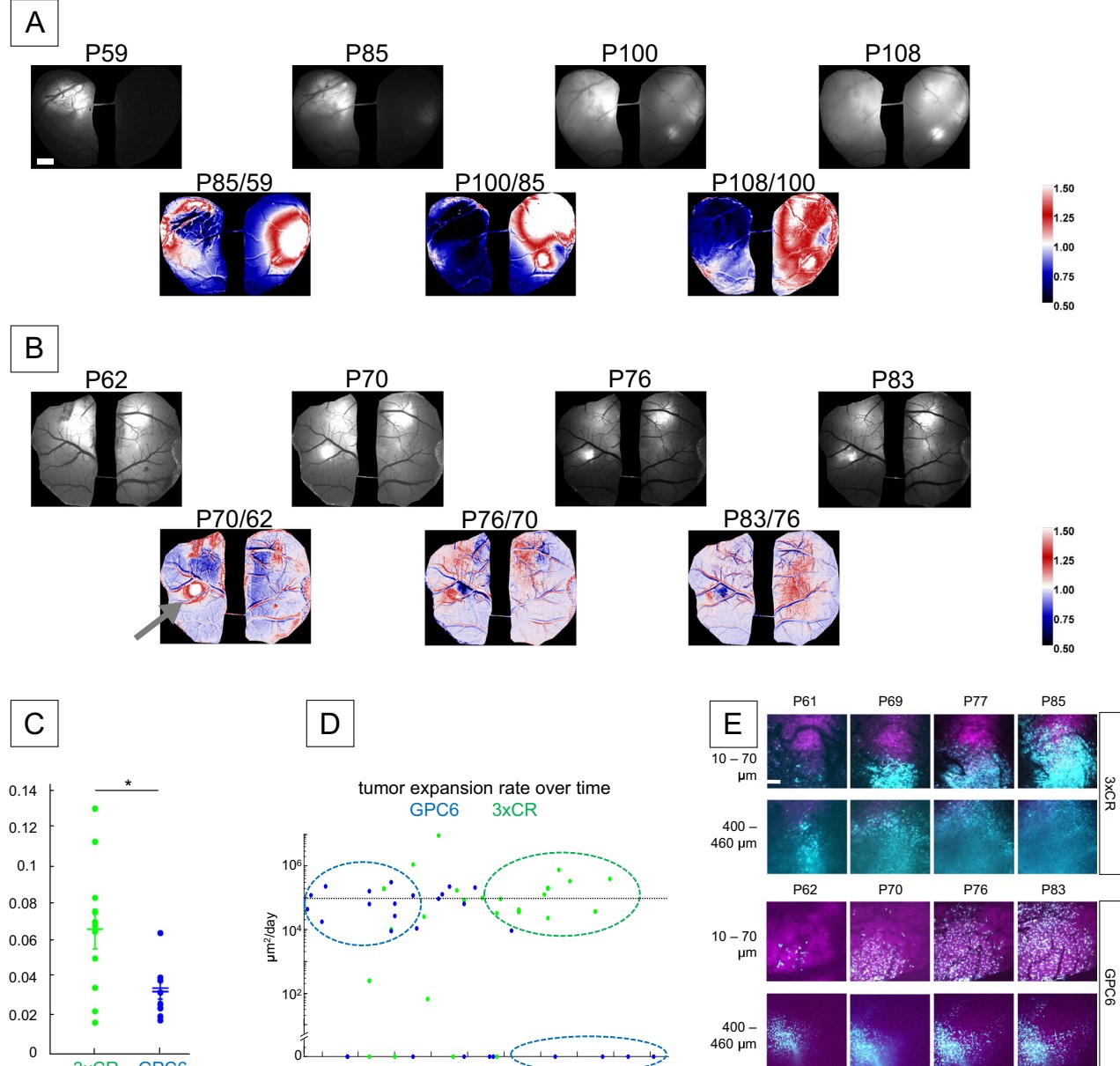

**Fig. 3 | GPC6 favors less rapid infiltration and earlier cortical tumor expansion.**
**A** Top row: representative mesoscopic images of tumor labeled fluorescence over time of a 3xCR tumor brain. Mouse age in postnatal days listed above image. Bottom row: colored panels, generated from above monochrome intensity, colored based on changes to signal strength at a location between two time points. **B** Analogous analysis to (**A**) for a GPC6 tumor animal. **C** Quantification of tumor images from 11 3xCR and 8 GPC6 comparing 90th percentile data point of CV/day. Mean 3xCR = 0.065 ± 0.011 sem. Mean GPC6 = 0.031 ± 0.005 sem, *p* = 0.033, two-tailed Wilcoxon rank-sum test. Asterisk denotes *P* < 0.05. **D** Mapping local tumor expansion rate by age in days. Expansion rates were calculated by changes in tumor fluorescence area over time (µm²/day). Blue and green dotted circles are designated to highlight relatively early expansion burst of GPC6 tumors, while 3xCR tumor expansion persists later in survival. **E** Representative images from two-photon imaging at different depths of tumor cells (cyan) and neurons (magenta). Visualized depth labeled on the left. Age at visualization labeled above images. Source data are provided as a Source Data file.

However, it does not permit cellular-resolution analysis, and the mesoscale signal is dominated by dendritic and axonal activity that may obscure the underlying soma voltage signal[33,34]. Therefore, we used awake two-photon high-resolution calcium imaging to analyze somatic glutamatergic neuronal activity patterns and ensemble metrics at the cellular level in seven 3xCR, six GPC6, and eight non-tumor control (supplementary Fig. 9) animals over time. First, we asked whether activity patterns were disrupted between intra- and extramarginal areas at early time points (see Fig. 6A for example recordings, 6D for neuron-tumor images). Overexpression of GPC6 caused a 48% higher intra−than extramarginal overall activity (dΔF/F) level, which was due to higher event rates and amplitudes

(*p* = 4.3e-7, *p* = 2.4e-9, respectively, KW/mc test). In 3xCR animals, activity was only qualitatively different between intra- and extramarginal sites with a lower event rate inside the margin (by 20%, *p* = 3.7e-5, KW/mc test; duration was significantly elevated, data not shown). Comparing across tumor genotypes, extramarginal activity was 71% higher in 3xCR animals (*p* = 1.2e-4, KW/mc test), which was due to an elevation in amplitude (by 17%, *p* = 0.0012, KW/mc test). 3xCR activity was 39% less clustered (i.e., organized in coactive neuronal ensembles; *p* = 0.014, KW/mc test). On the other hand, intramarginal events were 31% less frequent (*p* = 2.6e-13, KW/mc test) and 11% smaller (*p* = 4.55e-4, KW/mc test) in 3xCR brains, but of longer duration (data not shown).

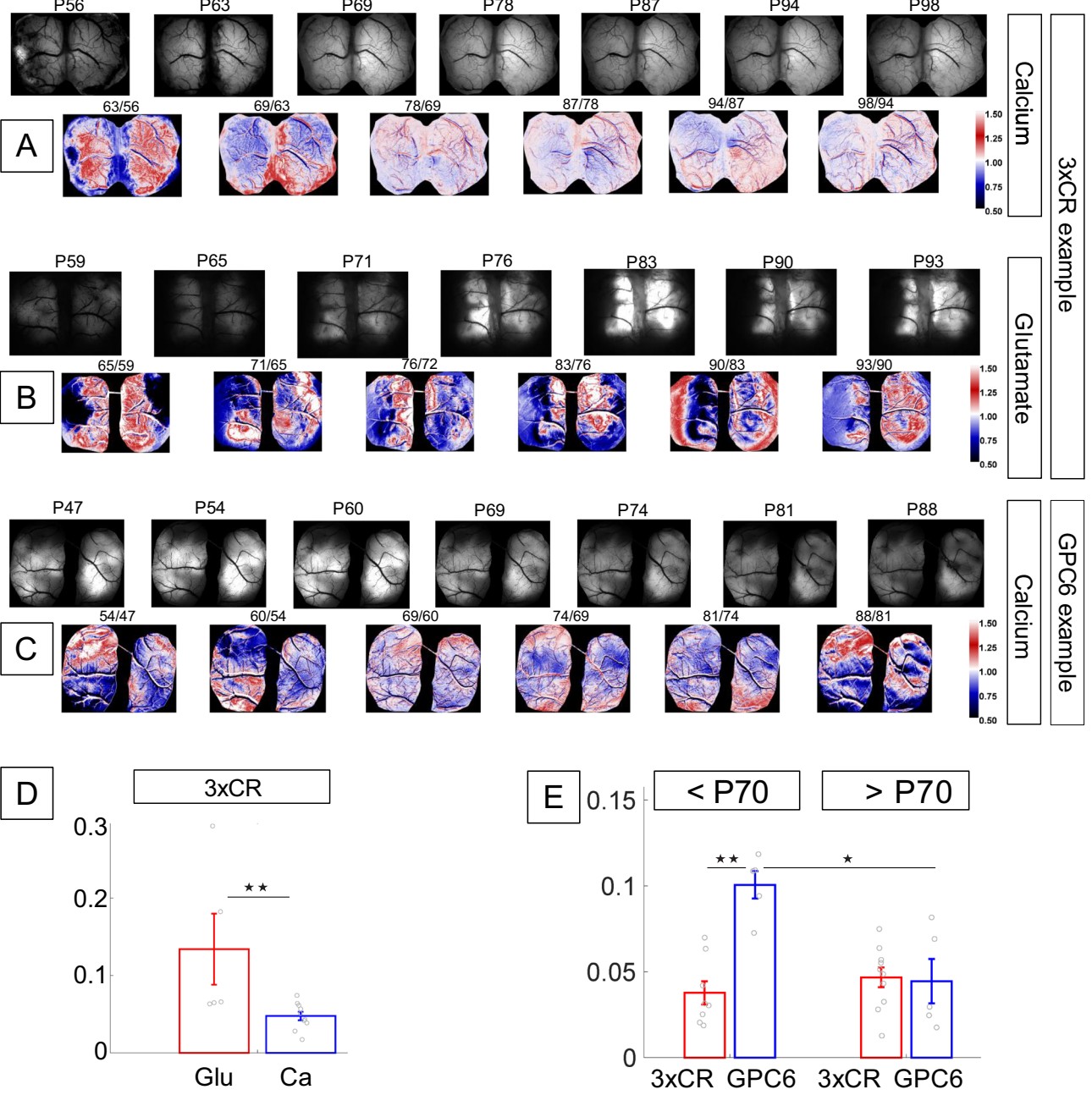

**Fig. 4 | Abnormal glutamate accumulation outpaces spatial extent and temporal increase in baseline neuronal calcium activity in 3xCR tumor cortex.** GPC6 baseline calcium signal is elevated at early time points (<P70). **A** Top: Black/white panels show calcium fluorescence of the widefield FOV in a 3xCR tumor cortex at seven time points between P56 and P98; equal brightness scale in each image. Bottom: Colored panels show the rate of change in tumor fluorescence at weekly intervals, generated by dividing two B/W images from consecutive time points, normalizing by the number of days between those sessions and by the mean of the resulting image, applying a color map (imageJ, "union jack") and setting the scale limits to [0.5 1.5]. **B** Analogous to (**A**), B/W panels show the evolution of baseline glutamate fluorescence intensity in a different 3xCR tumor animal between P59 and P93 with equal brightness scaling. Note the distinct nonlinear, progressive changes in signal intensity not seen in the calcium-reporter mouse (**A**). Colored panels were constructed and scaled as in (**A**). **C** Analogous to (**A**), B/W panels show calcium baseline signal snapshots from a GPC6 tumor animal between P47 and P88, as well as the CV/day fluorescence change between time points. **D** To

capture the dynamical changes in fluorescence between time points, we computed the coefficient of variation (CV = SD/mean), normalized by the number of days between recordings. To account for potential sampling bias/undersampling, we extrapolated a 90th percentile data point for each animal. Data were shown from ten 3xCR calcium-reporter mice and five 3xCR glutamate-reporter mice. Glutamate CV/day reached 64% higher values than calcium across all time points (mean $0.134 \pm 0.046$ sem vs. $0.048 \pm 0.0055$ sem, $p = 0.008$, two-tailed Wilcoxon rank-sum test). **E** Comparison of calcium baseline fluorescence between <P70 and >P70 time periods in 3xCR (data from ten animals) vs. GPC6 (data from five animals) tumors: GPC6 calcium baseline (<P70) was higher than 3xCR (<P70), and both GPC6 and 3xCR at P > 70: mean GPC6 calcium <P70/> P70: $0.1 \pm 0.008$ sem/ $0.047 \pm 0.006$ sem, mean 3xCR calcium <P70/>P70: $0.038 \pm 0.007$ sem/ $0.045 \pm 0.013$ sem; comparison <P70 3xCR vs GPC6: $p = 0.0064$; comparison GPC6 < P70 vs >P70: $p = 0.04$, KW/mc test). Source data are provided as a Source Data file. In **D**, **E**, single asterisk denotes $P < 0.05$ and double asterisk denotes $P < 0.01$.

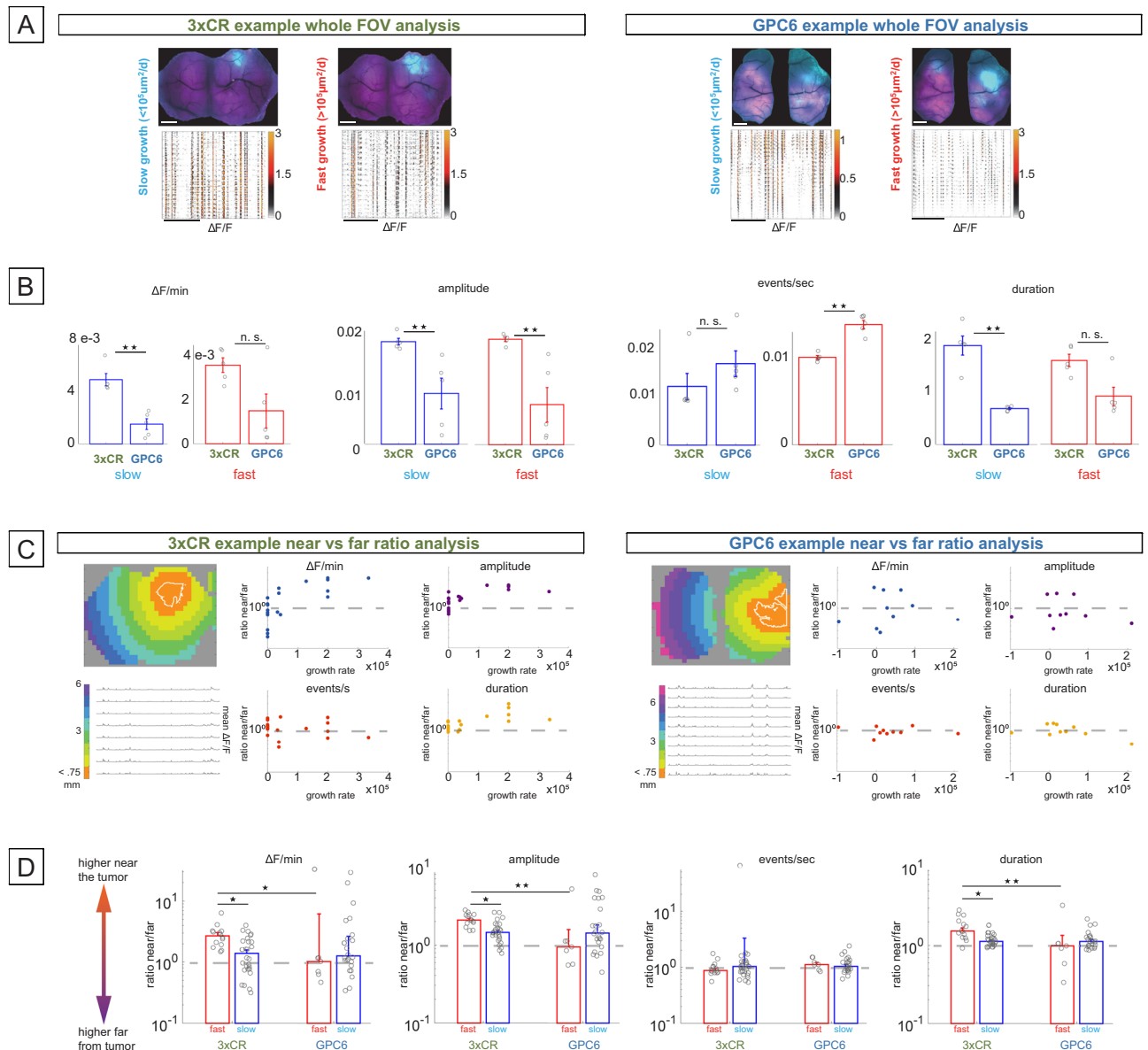

**Fig. 5 | Mesoscopic neural activity patterns are significantly lower with GPC6 overexpression when the local tumor expansion rate is below $10^5\ \mu m^2/day$; Neural activity patterns drop off faster with distance from 3xCR tumors when proliferation rates are high than when they are low; GPC6 tumors generally do not induce this effect. A** Left: Example of 3xCR whole-FOVs values for two time points corresponding to slow ($3.3 \times 10^4\ \mu m^2/day$, left) and fast ($2 \times 10^5\ \mu m^2/day$) tumor expansion rate (right). Top panels: tumor (cyan) and calcium (magenta) FOV of the same animal at different time points. Horizontal scale = 1 mm. Bottom: representative filtered and denoised ongoing calcium traces, one row per pixel (after spatial downsampling, 520 pixels total, color scale: 0 to $3 \times 10^{-3}\ \Delta F/F$).). Scale = 50 s. Right: As in **A** (left), the top panels show tumor (cyan) and calcium (magenta) fluorescence, and the bottom panels spontaneous $\Delta F/F$ calcium activity of a GPC6 tumor animal from representative recordings at a slow tumor expansion stage (left), and a fast expansion stage (right). **B** Comparative bar plots for four activity metrics ($\Delta F/min$, events/sec, amplitude, duration), including pooled data from five 3xCR animals (24 recordings during slow expansion, 18 recordings during fast expansion) and five GPC6 animals (20 recordings during slow tumor growth, 13 recordings during fast expansion); blue bars = slow expansion recordings, red bars = fast expansion recordings. Error bars = standard error of the mean. Star denotes a significant difference between fast and slow expansion

conditions, n.s. not significant. **C** Left: Example of a 3xCR tumor recording analyzed by distance from the tumor edge over time. Top left panel: The distance from the tumor edge was computed. Distance bands are color-coded in 0.75 mm increments. Bottom left: Mean $\Delta F/F$ traces: 30-s period of mean quiet spontaneous activity corresponding to the adjacent distance-band color scale. Scatter plots to the right: ratios between distance bands <0.75 mm from the tumor edge versus >3 mm, averaged over all pixels inside the respective distance bands, as a function of expansion rate. One distinct recording per data point. Y-axes are at a logarithmic scale to visualize ratios symmetrically around the $10^0$ point (ratio of 1, dashed gray line). Right: data from a GPC6 tumor animal, analogous to the 3xCR example. **D** Results from five 3xCR (27 recordings during slow tumor expansion, 14 recordings during fast expansion) and five GPC6 animals (24 recordings during slow expansion, seven recordings during fast expansion). As in **C**, values above the $10^0$ line represent instances of metrics higher near the tumor edge than far away, and below vice versa. Error bars = standard error of the mean. Significantly different comparisons are only shown between periods of fast and slow expansion within tumor genotypes and within fast or slow conditions across genotypes. Source data are provided as a Source Data file. In **B**, **D**, the single asterisk denotes $P < 0.05$ and the double asterisk denotes $P < 0.01$.

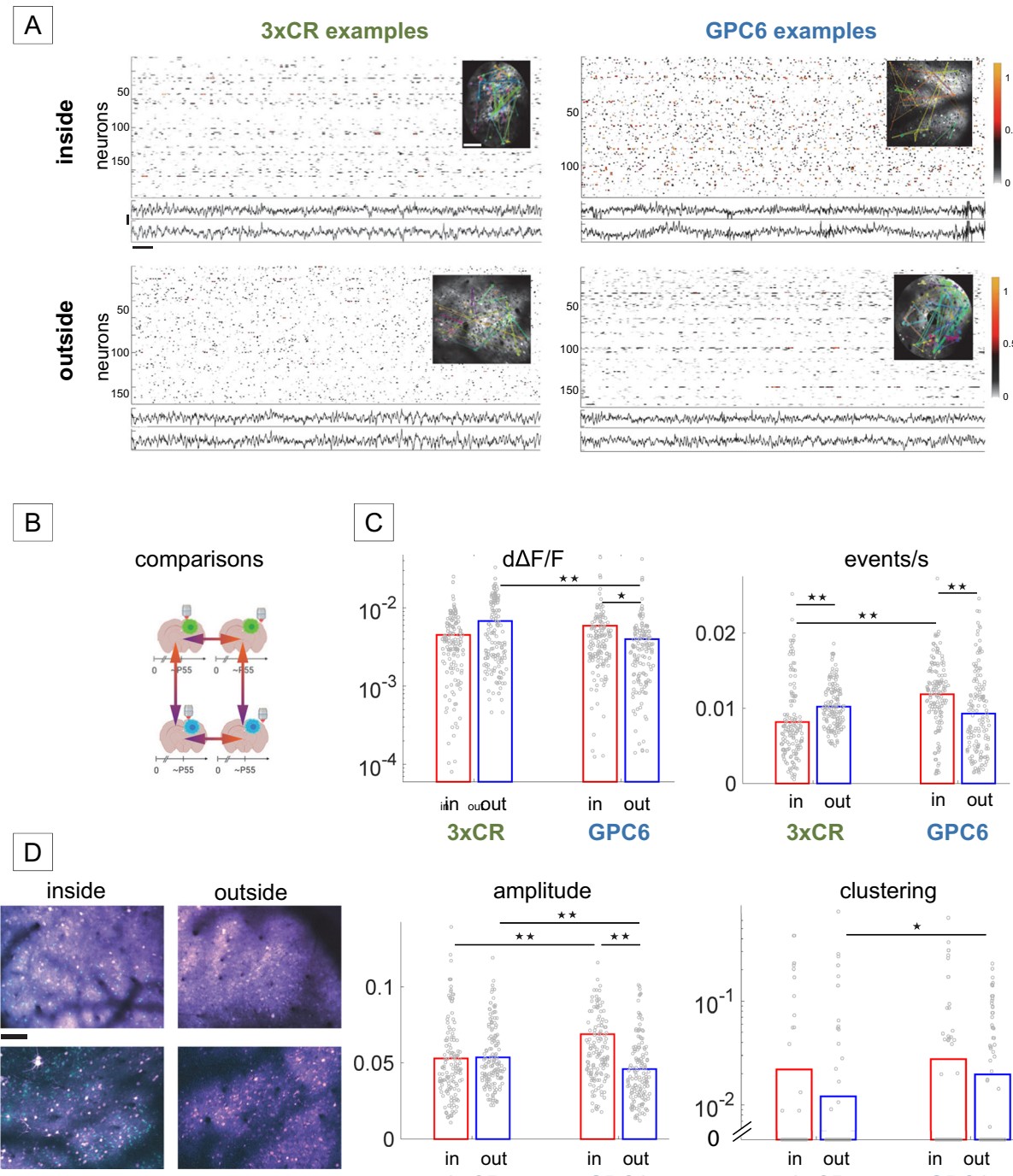

**Fig. 6 | 3xCR and GPC6 alter neuronal activity inside and outside the tumor margins in distinct ways. A** Comparison of calcium activity inside tumor margins (top two panels) vs outside (bottom two panels), for 3xCR tumors (left) and GPC6 tumors (right). Each panel consists of a raster plot of deconvolved calcium activity (one row per neuron), 2-ch EEG (vertical scale = 200 μV), and an insert showing the FOV with clustered neurons in matching colors and line connecting pairs of neurons within the clusters (line width proportional to connection strength). The "3xCR inside" and "GPC6 outside" recordings were acquired in spiral scan mode, whereas the "3xCR outside" and "GPC6 inside" FOV's were imaged under resonant scan. Horizontal scale = 1 s. **B** Schematic visualization of meaningful comparisons highlighted in this figure between 3xCR intramarginal vs extramarginal FOVs, between GPC6 intra- and extramarginal FOVs, and across genotypes within intra- or extramarginal locations. Significant differences between these groups are marked with horizontal bars in (**C**). The brain section pictograms indicate tumor type

(3xCR=green, GPC6 = blue), time bins, and imaging locations. Created with BioRender.com **C** Comparison of activity metrics between neurons inside and outside the tumor margin derived from five 3xCR and six GPC6 tumor animals. Only significant changes for mean deconvolved ΔF/F, mean events/sec, mean event amplitude, and mean clustering coefficients are highlighted by horizontal lines above the bar (mean and individual data points, 150 data points per group, each point corresponding to one neuron) graphs. We do not show the duration metric as this information is contained in, and can be inferred from, the combination of dΔF/F, event rate, and amplitude. The clustering coefficient and dΔF/F data points are plotted on a logarithmic y-axis to account for the high variability in measured values. Single asterisk denotes $P < 0.05$, double asterisk denotes $P < 0.01$. **D** Example of GPC6 FOV's inside and outside the tumor margin (top), analogous for Cr86 tumors (bottom). Tumor cells = cyan, neurons = magenta. Scale bar = 100 μm. Source data are provided as a Source Data file.

Unexpectedly, this result exposed a significant difference in somatic activity patterns inside vs. outside the tumor margin, with a strong dependence on tumor genotype, and with a drop-off with distance in GPC6 animals, not 3xCR. Activity inside the GPC6 tumor consisted of high-amplitude, high-frequency, short events, with frequency and amplitude reducing towards distal areas. Activity inside 3xCR tumor margins had low event rates and long durations, changing towards more frequent and shorter bursts going outward from the tumor.

The datasets used here were taken at somewhat earlier time points than the widefield imaging data (Fig. 5): 3xCR inside mean age (day P) = $57.6 \pm 4.8$ SD, GPC6 inside: $49.8 \pm 7.9$ SD, outside 3xCR: $54.2 \pm 2.5$ SD, outside GPC: $53.8 \pm 2.3$ SD, control: $53 \pm 6.1$ SD. The previous mesoscopic analysis (Fig. 5) taken during periods of rapid cortical tumor expansion were at a mean age of $96.3 \pm 23$ SD (3xCR), and $77 \pm 17.5$ SD (GPC6), on average at least 20 days later. This agrees

with earlier observations of 3xCR tumors showing more enduring activity than their GPC6 counterparts (Fig. 3).

## Dynamic changes in neurons outside the leading edge reflect GPC6 expression

To probe the range of tumor influence analogously to the expansion dynamics analysis (Fig. 3), we asked whether neurons located 1–2 mm beyond the tumor margins are affected differentially over time depending on tumor genotype, as assessed by single-unit metrics and network cluster analysis[35–37]. Figure 7 shows all meaningful comparisons across tumor genotypes and time points for the four metrics dΔF/F, event rate, amplitude, and clustering coefficient. Note that dΔF/F and clustering coefficient plots use a logarithmic y-axis to more conveniently display the wide distribution of individual cell data points. In 3xCR tumor animals, cellular mean dΔF/F activity was, on average, 40% higher at mid vs. early time points ($p = 0.013$, KW/mc test,

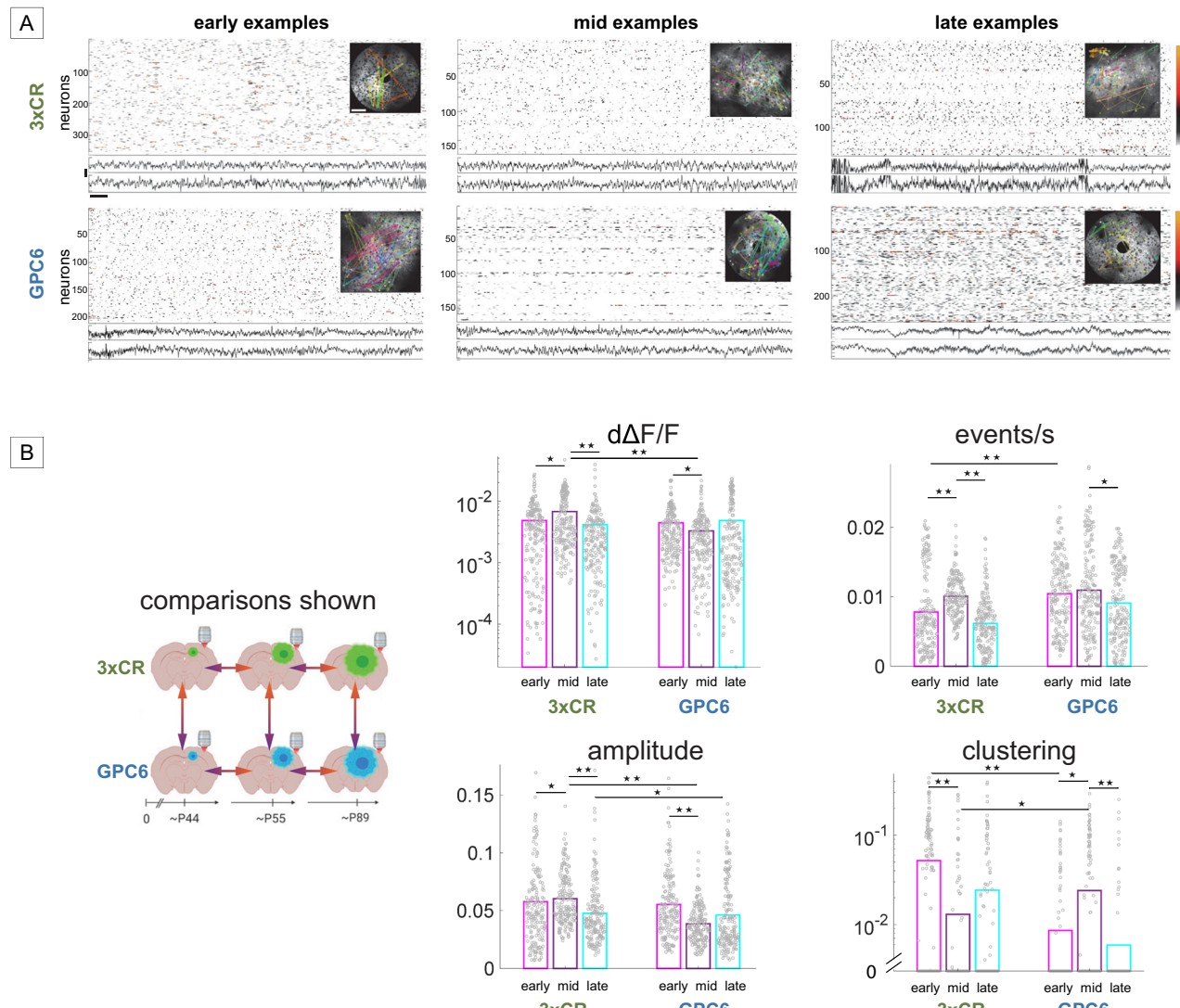

**Fig. 7 | Distinct temporal dynamics of 3xCR and GPC6-induced changes in neuronal activity patterns located 1–2 mm beyond the tumor margin.**
**A** Example plots of dΔF/F, EEG, and clustering for early (P41–49, mid (P49–56 and late (P68–129) time bins, analogous to Fig. 6A: dΔF/F, EEG's (vertical scale = 200 µV, horizontal scale = 1 s), and FOV insert with neuronal clusters (white scale bar = 100 µm. "3xCR early", "GPC6 mid", and "GPC6 late" examples were acquired in spiral scan mode, the rest in resonant scan mode. **B** Left: schematic of comparisons between the two tumor genotypes and three time periods presented in this figure,

analogous to Fig. 6B. Created with BioRender.com Right: Comparison of activity metrics across three time bins for both tumor genotypes: Analogous to Fig. 6, bar plots show mean neuronal deconvolved dΔF/F activity, mean event rates, mean calcium transient amplitudes, and clustering coefficients across time points and tumor genotypes. Individual data points correspond to single neurons (200 per group). Data were derived from seven 3xCR and six GPC6 tumor animals. Single asterisk denotes $P < 0.05$, double asterisk denotes $P < 0.01$. Source data are provided as a Source Data file.

Fig. 7B), and event rate and amplitude were elevated as well, however, clustering coefficients were significantly lower by 75% ($p$ = 4.3e-10, KW/mc test), indicating disorganized, less burst-dominated activity patterns with much shorter events, and pointing towards a dysregulation of local functional ensemble structure starting between early and mid time periods.

In GPC6 animals, on the other hand, cellular d$\Delta$F/F activity was 26% lower at mid time points compared to early ($p$ = 0.04, KW/mc test), and mean amplitude was reduced by 32% ($p$ = 3.6e-8, KW/mc test), indicating lower firing rates inside bursts (event rates were unchanged), and increased functional clustering compared to the early stage. By contrast, from mid to late time points, overall activity was unchanged but the mean rate of Ca$^{2+}$ events/sec was 26% lower ($p$ = 0.04, KW/mc test), and mean CC decreased by 75% ($p$ = 2e-5, KW/mc test).

These results indicate that both tumor types alter extramarginal neural network activity over time but with different temporal dynamics. Specifically, 3xCR tumors caused significant disorganization of ensemble activity patterns from early towards mid and late stages, whereas GPC6 tumors early on showed elevated firing and amplitude (indicating intra-burst frequency) as well as an early reduction in functional organization.

Lastly, we directly compared the two tumor genotypes within the 3 time bins to quantify differences in their impact on extramarginal neuronal activity parameters during progressive stages of tumor invasion (Fig. 7B, 3xCR vs. GPC6 comparisons). At early time points, the mean CC of neurons around GPC6 tumors was 83% lower than around 3xCR tumors ($p$ = 1.9e-10, KW/mc test), but the event rate was 34% higher ($p$ = 2.7e-8, KW/mc test). At mid time points, mean d$\Delta$F/F was 52% lower around GPC6 tumors than around 3xCR tumors, ($p$ = 2e-8, KW/mc test), and mean amplitudes were 29% lower ($p$ = 5e-11, KW/mc test), suggesting overall less activity within neuronal populations around GPC6 tumors. At late time stages, amplitudes were similarly reduced by 11% ($p$ = 0.024, KW/mc test). Taken together, differences between GPC6 and 3xCR peritumoral neuronal patterns underwent several transitions over time, from more frequent but less organized bursts in GPC6 tumors at the early stage, to less activity with an increased functional organization at the mid stage, followed by overall lower amplitude and frequency but unchanged d$\Delta$F/F at late time points compared with 3xCR tumors.

## Discussion

For many decades, GBM has consistently ranked as one of the least successfully treated human malignancies, with the highest recurrence and lowest survival rates, despite the introduction of promising therapeutic interventions[21]. In addition to poor survival prognoses, GBM routinely occurs with severe neurological comorbidities, including early seizure onset and cognitive deficits[2,38], suggesting a functional interrelationship. A spectrum of mechanisms underlying this hyperexcitability[22,39–42] has been proposed, supported by several studies indicating that anti-epileptic drugs may slow tumor progression[1,43–45].

Despite these advances, many questions about the evolution of reciprocal feedback mechanisms and their spatial reach between GBM and the surrounding neural activity remain unanswered. Here we use a recently established native GBM model in an immunocompetent host[4,19], a model that recapitulates many hallmarks of the human disease[8], to chronically visualize and correlate glutamate, neuronal hyperactivity, and tumor expansion in vivo at high temporal resolution. We show evidence of early extensive cortical glutamate accumulation as 3xCR tumor cells invade the cortex that is not accompanied by equally dramatic changes in baseline calcium signaling, suggesting that it does not depend exclusively on synaptic release. This finding supports ample evidence of multiple other molecular pathways that can alter the GBM extracellular microenvironment by raising free glutamate levels[23,42,46–48].

GBM invasiveness rates vary significantly in humans, and we replicated this phenomenon in our finding that local cortical tumor infiltration dynamics were not sustained into late stages (as opposed to the 3xCR model, Fig. 3) when we added overexpression of a single glypican gene linked to synaptogenesis (GPC6) to the IUE CRISPR construct. Likewise, we saw an early surge in baseline aggregate (mesoscopic) calcium signal not seen in 3xCR animals (Fig. 4). Glutamate release dynamics appeared to be uncoupled from baseline mesoscale calcium levels in 3xCR tumor brain. A possible explanation for this unexpected result could be that chronic exposure to excess glutamate dampens AP output and synaptic activity over time via neurotoxic and degenerative mechanisms[49,50], however a detailed analysis of the underlying mediators in our model will need to be undertaken in future studies. Novel glutamate and voltage sensors with improved signal characteristics for cellular-resolution imaging will reveal the relationship between chronic glutamate exposure and changes in neural excitability in greater spatial and temporal detail. GPC6 tumors presumably alter glutamate signaling significantly earlier than 3xCR, however we were unable to obtain measurements before P45, leaving it up to future studies to determine whether GPC6 addition may impact glutamate accumulation at very early disease stages. Furthermore, we established that distinct cellular neuronal network activity disruption, both between intramarginal and extramarginal neuronal populations and over time at locations >1 mm beyond the tumor edge, depends on the genetic makeup of the tumor. Recent work has implicated several subtypes of glypicans, the astrocytically secreted proteins that promote glutamate receptor expression and induce functional synapse formation, and GPC6 was linked in particular to excitatory synapse growth[25–27]. We previously showed that GPC3, a member of the glypican family, can drive increased peritumoral neosynaptogenesis when expressed in GBM using our IUE CRISPR/Cas9 model[19]. While it is possible that the GPC6-induced peritumoral neosynaptogenesis at P30 (Fig. 2G, H) underlies the early infiltration and calcium dynamics we observed, further analysis will be needed to fully understand the disparate contributions of GPC homologs to tumorigenesis and their timing throughout disease progression.

Analyzing the mesoscopic calcium activity patterns in both 3xCR and GPC6 animals revealed that multiple activity metrics were significantly shifted during periods of high versus low cortical tumor expansion rates, depending on tumor genotype. The activity was generally greater in 3xCR brains than in GPC6 when tumor expansion was slow, and when expansion accelerated, GPC6 brains favored higher event rates with lower amplitude, indicating unexpected disturbances in mesoscale activity patterns. At a finer spatial scale, we found that 3xCR tumor animals showed a stronger difference in calcium activity profiles between periods of fast and slow tumor spread: calcium events were larger close to the tumor edge during fast tumor expansion, whereas in GPC6 brains, we did not see an equally significant distance relationship with activity metrics. On a cellular level, we compared glutamatergic cortical neurons across time, tumor distance, and genotype. 3xCR tumors caused increased neuronal firing compared to GPC6 outside the tumor margin, but neurons inside the GPC6 margin were more active than those outside. 3xCR peritumoral neurons also showed early-to-mid point reduction in somatic co-activity clustering with more frequent bursts. Compared to GPC6 brains, 3xCR activity levels were higher at mid and late time points, underscoring the significant differences in temporal dynamics of tumor/neuron crosstalk at the leading edge. Building on these results (see supplementary Fig. 10 for a schematic summary), we anticipate that future studies will yield additional valuable characterization of epileptogenic processes, such as a detailed analysis of changes in the balance of excitation and inhibition, as a function of tumor infiltration and tumor genotype.

These results indicate an unexplained but important difference in the way the specific genetic makeup of GBM may influence how it interacts with the surrounding microenvironment. Our current understanding that tumor genetics drive the progression of the cortical hyperexcitability microenvironment and hence impairment of higher order cortical function predicts that matching precision GBM diagnoses with information on residual disease can help guide future tailored management of the neurological comorbidity of brain tumors.

## Methods

### Animals

All animal procedures were carried out under an animal protocol approved by the Baylor College of Medicine IACUC. Mice were housed at 21 °C under a 12 h/12 h/ on/off light cycle, with humidity set at 45%. IUE CRISPR mice were generated on CD-1 and C57BL/6 backgrounds as previously described[4,8,19] (see below). We used WT CD-1 mice for AAV-GCaMP, jrGeco1a, and iGlusnfr injections, or thy1-GCaMP6s-4.3 expressing mice on a mixed CD-1 x C57-BL6/J background. Animals of both sexes were used in the study; sex was not a factor in the experimental design and was not considered in random animal assignments to experimental groups. Sex information is included in Supplementary Table 1, listing all animals and in which figures data from each animal appear.

### In utero electroporation (IUE) model of GBM

In utero electroporation was performed on embryonic day 14.5, as described previously[51]. After pregnant moms were anesthetized, the uterus was exposed, and a plasmid cocktail was injected into the lateral ventricles of developing embryos. Progenitor cells lining the lateral ventricles were electrotransfected, after which the uterus was reinserted, and surgical openings were closed. The injection mixture included a single construct which expressed previously reported guide sequences to target Pten, Trp53, and Nf1, along with the Cas9[4,15,19]. A piggyBAC transposable system was utilized to label cells with a GFP reporter and/or overexpressed GPC6. The CRISPR guide sequences and the model were previously published (PMID: 28166219). When we developed the model, the guide sequences were adopted from previously validated studies (PMID: 25119044 and 24952903). The specific target sequences of the guides are as follows (in order, Tp53, Pten, and Nf1): CCTCGAGCTCCCTCTGAGCC, GAGATCGTTAGCAGAAACAAA, and GCAGATGAGCCGCCACATCGA.

### Immunohistofluorescence analysis

After IUE, mice were born and aged to postnatal day 30, at which point mice were euthanized and brains were dissected after transcardial perfusion of phosphate-buffered saline (PBS) followed by 4% paraformaldehyde. After a brief post-fix, tissues were cryoprotected in 30% sucrose overnight. Brains were then embedded in the Tissue-Tek OCT Compound (Sakura Finetek USA, 4583). Brains were sectioned at 40 μm thickness and processed with antigen retrieval for their associated antibody staining. For BrdU-cell proliferation analysis, 2 h before collection, 100 μg BrdU (in PBS) per gram body mass were delivered through intraperitoneal injection. Mouse brains were collected, frozen, and sectioned as described above. Before blocking, sections were incubated in 2 N HCl at 37 °C for 30 min and neutralized with 3.8% sodium borate for 10 min at room temperature. For synaptic staining, sections were treated in 10 mM sodium citrate, 05% Tween 20, pH 6.0 at 75 °C for 10 min.

The following primary antibodies were used (at their associated dilutions, including catalog number (cn), and lot number(ln)): rat anti-BrdU (BU1/75 (ICR1), 1:200; abcam, cn: ab6326, ln: GR3269246-1), mouse anti-gephyrin (1:500; Synaptic Systems, cn: 147011, ln: 1-64), rabbit anti-GFP (1:1,000; Thermo Fisher, A-11122), mouse anti-PSD95 (7E3-1B8, 1:500; Thermo Fisher, cn: MA1-046, ln: LI147875), guinea-pig anti-VGAT (1:500; Synaptic Systems, cn: 131004, ln: 2-41), guinea-pig

anti-VGLUT1 (1:2,000; Millipore, cn: AB5905, ln: 3193844). We used species-specific secondary antibodies tagged with Alexa Fluor 488, 568, or 647 (1:1,000, Thermo Fisher) for immunofluorescence. After Hoechst nuclear counterstaining (Thermo Fisher, H3570, 1:50,000), coverslips were mounted with VECTASHIELD antifade mounting medium (Vector Laboratories, H-1000).

### Quantification of peritumoral synapses

Excitatory and inhibitory synapses at the tumor margins in cortical regions were quantified as previously published[19,28,29,52]. Briefly, frozen brains were sectioned to 40-μm thickness and stained with the aforementioned pre- and post-synaptic markers for excitatory (VGlut1/PSD95) and inhibitory (Vgat/Gephyrin) synapses. Confocal images were taken with a Zeiss LSM 880, and functional synapses were identified by the colocalization of the pre-/post- pair using the Synapse-Counter plugin in ImageJ[53]. We used a control section which was stained with only secondary antibodies to establish threshold levels for discerning positive staining compared to no background fluorescence.

### Human glioma tissue analysis

Human glioma tissue microarrays were provided by Baylor College of Medicine's Pathology and Histology Core. Sections were deparaffinized, rehydrated, and prepared for GPC6 staining (goat anti-mouse Gpc6, 10 μg/mL, R&D Systems, cn: AF1053, ln:GJA0219031) through the following treatment: 3 × 3 min in xylene, 3 × 3 min in 100% ethanol, 3 × 3 min in 95% ethanol, 3 min in 80% ethanol, 5 min in 70% ethanol, 5 min in 50% ethanol, 5 min in ddH$_2$O, antigen retrieval (10 mM sodium citrate, 0.05% Tween 20, pH 6.0) at 95 °C for 5 min, cool to room temp, 30 min in 3% hydrogen peroxide, 30 min in 0.1% Triton X-100 in PBS, and 30 min blocking with normal horse serum (provided in Vector Lab ImmPRESS HRP Horse Anti-Goat Polymer Detection Kit, MP-7405). After primary and secondary staining, a signal was developed using the DAB Substrate Kit (Vector Lab, SK-1000) supplement with nickel. Sections were treated with Harris hematoxylin for 15 s and the staining was blued with running tap water. Sections were then dehydrated and sealed with Permount before imaging.

### Survival analysis

After mice were born, they were observed for symptoms suggestive of tumors, including (but not limited to) lethargy, hunched posture, decreased appetite, decreased grooming, trembling/shaking, squinting eyes, partial limb paralysis, and abnormal gait, denoting the IACUC permitted endpoint. Statistical analysis was performed using the ggsurplot() function of survminer package (v0.4.9).

### RNA-Sequencing and bioinformatics analysis

Samples from tumor brains were collected from an endpoint, where tumor tissue was collected with the aid of fluorescence microscopy. Total RNA was isolated using the RNeasy Plus Mini Kit (Qiagen, 74134) according to the manufacturer's protocol. Samples were prepared in biological replicates, $n \geq 3$ per variant genotype. RNA integrity (RNA integrity number $\geq 8.0$) was confirmed using the High Sensitivity RNA Analysis kit (AATI, DNF-472-0500) on a 12-Capillary Fragment Analyzer. Illumina sequencing libraries with 6-bp single indices were constructed from 1 μg total RNA using the TruSeq Stranded mRNA LT kit (Illumina, RS-122- 2101). The resulting library was validated using the Standard Sensitivity NGS Fragment Analysis Kit (AATI, DNF-473-0500) on a 12-Capillary Fragment Analyzer. Equal concentrations (2 nM) of libraries were pooled and subjected to sequencing of ~20 million reads per sample using the Mid Output v2 kit (Illumina, FC-404-2001) on an Illumina NextSeq550 following the manufacturer's instructions.

FASTQ files were quality controlled using fastQC (v0.10.1) and MultiQC (v0.9),11 and aligned to the mm10 reference genome via STAR (v2.5.0a).12 Count matrices and gene models were built from aligned files using Rsamtools (v2.0.0) and GenomicFeatures (v1.32.2) in

R (v3.5.2). Using DESeq2 (v1.20.0) samples were normalized and analyzed for differential gene expression where GPC6 tumors were compared to 3xCR. Genes were considered significantly differentially expressed with a fold change >±1.5 at $P < 0.01$. Ontology analysis was performed with the assistance of the Enrichr tool (maayanlab.cloud/Enrichr/).

Human expression and survival correlation analysis was performed utilizing the GEPIA v2 online tool. Low and high expression cohorts were set at the bottom and top 50%, respectively. Log2 fold change cutoff was set at 1, $P$ value cutoff was set at 0.01, and match normal data were set to TCGA normal and GTEx data.

## EEG and headpost implant, virus injection, craniotomy

Mice were induced with 3–4% isoflurane in oxygen at 2 L/min flow rate until breathing slowed to ~60 bps. Deep anesthesia was confirmed via tail pinch. Hair was removed by electric clippers (Wahl, Sterling, IL, USA), and depilatory cream (Nair, Church & Dwight, Ewing, NJ, USA). The mouse was immobilized in a stereotactic frame with a gas-anesthesia adapter for neonatal rats and blunt-tip ear bars aimed at the upper mandible joint (Kopf Instruments, Tujunga, CA, USA). The scalp was disinfected using three alternating applications of betadine scrub and 70% EtOH, and the animal was covered with a fenestrated sterile drape, sparing only the disinfected scalp. All surgical instruments were autoclaved prior to use. Using pointed-tip surgical scissors (WPI, Sarasota, FL, USA), a ~1 × 1 cm square skin flap was removed to expose the skull bone. Using a no. 10 scalpel blade and serrated forceps, the periosteum was removed, and residual bleeding was absorbed with sterile cotton swabs. A thin layer of Vetbond cyanoacrylate (3 M, Maplewood, MN, USA) was applied to close the scalp wound. Next, a custom-made EEG adapter with attached wires (two recording channels, one reference, each made from Teflon-coated silver (WPI, Sarasota, FL, USA) wire with ~0.5 mm tips exposed, beveled using 600 grit sandpaper and chloride in NaClO) was attached using Krazy glue (Elmer's, Westerville, OH, USA) to the skull bone ~3 mm anterior of lambda. A dental drill with ¼ carbide burr tip (Midwestern, Henry Schein, Melville, NY, USA) was used to make 3 burr holes, one over the cerebellum (bregma ~3 mm, 2.5 mm lateral), and two at lambda +1 mm, 5 mm lateral on both hemispheres, making sure not to break through the bone but to create small cracks exposing the underlying dura mater. The silver wire tips were carefully inserted just below the bone to touch the dura, and the holes were immediately sealed with a small drop of Vetbond. A mixture of dental cement powder and charcoal powder (10:1) was mixed with dental cement fluid (Lang dental, Wheeling, IL, USA), and applied to the skull in a donut shape, approx. 5 mm tall and creating a ~15 mm diameter well. The custom-designed aluminum head bar (emachineshop, Mahwah, NJ, USA) was gently pushed into the dental cement to sit flush with the skull. A small amount of Vetbond was added to the inside walls of the headpost to promote permanent attachment. After the dental cement cured, excess cement was removed using the dental drill. A nanoliter injector (Nanoject II, Drummond scientific, Broomall, USA), was used to inject AAV solutions. Glass pipettes were pulled using a Sutter P-87 horizontal pipette puller (Sutter Instruments, Novato, CA, USA), and tips were broken on the filament of a vertical puller (Narishige, Japan) to create a sharp, ~10–20 μm wide opening. The pipette was backfilled with corn oil, and ~5 μL virus solution was aspirated from a sterile piece of parafilm using the Nanoject II. Depending on the experimental paradigm, we used a mouse line with genetically expressed calcium indicator (thy1-GCaMP6s_GP4.3, for expression characterization see ref. 54), or one of the following (combinations of) AAV's:

pGP-AAV-syn-jGCaMP7f-WPRE (Addgene, Watertown, USA; see[55,56] for pan-neuronal expression characteristics of AAV's under the h-synapsin promoter), diluted 1:2.5 in sterile Cortex buffer (CB),

or AAV-FLEX-GCaMP8m + AAV-CamKIIa-0.4-Cre (expression almost exclusively in pyramidal cells[57]), diluted 1:1.5 in sterile CB

or AAV-Syn-NES-jrGeco1a-WPRE-SV40, diluted 1:3 with sterile CB,

or AAV-FLEX-iGlusnfr-184 + AAV-Ef-1α-Cre[56], diluted 1:1.5 in sterile CB,

or AAV-FLEX-iGlusnfr-184 + AAV-CamKIIa-0.4-Cre, diluted 1:1.5 in sterile CB. See supplementary Table 1 for a complete list of animals and their respective fluorescent activity reporter types. For two-photon imaging, we only used animals with calcium reporter expressed in pyramidal cells, because we had previously shown that PV-expressing interneuron health inside the tumor margin is disproportionally degraded, and therefore it would have been problematic to compare all neurons inside and outside the tumor margin with each other. Based on the uniform delay between injection and expression level we observed inside the tumor margins as well as further away from the tumor, we are confident that there is no significant difference in the AAV uptake and intracellular action between intra- and extramarginal neurons. We did not observe uptake of AAV-mediated fluorescent reporters in tumor cells, however we cannot absolutely rule out that spurious uptake did occur in rare cases.

A total of eight or six (depending on the unobstructed skull area available) equidistant injection locations were selected with the goal to maximize coverage of the sensory and posterior motor cortex. For each hemisphere, these were at [bregma −3.5 mm; 2.5 mm lateral], [bregma −2 mm; 1 mm lateral], [bregma −2 mm; 4 mm lateral], and [bregma −0.5 mm; 2.5 mm lateral]. At each location, a total of 400 nL AAV-solution at two depths, ~300 and ~600 μm, were injected into the cortex in 9.2 nL/pulse increments separated by 10 s. After the last pulse in each location, the pipette remained in place for 5 min to minimize backflow and promote virus diffusion. The Nanoject was mounted at a 25-degree angle relative to the skull surface at each location and actuated by a manual and 1-direction motorized micromanipulator (WPI, Sarasota, FL, USA) at speeds of ~600 μm per min. After the last injection, the skull was covered with Vetbond and dental cement. Mice were allowed to recover for ~10 days post injection with meloxicam analgesic administered for the first 3 days. For the second surgery, induction and stereotactic positioning were performed as described above. The dental cement covering the skull was removed. A single-pane #1 cover glass (22 × 11 mm, Thomas Scientific, Swedesboro, NJ) was cut and adjusted to the individual dimensions (~7.5 × 6 mm) of the exposed skull bone by scoring the glass with a sharp-tip drill bit. The cover glass was then submerged in 70% EtOH for >10 min. The outline of the cut glass was transferred, and a corresponding groove was carefully drilled while cooling the bone every 10–20 s with compressed air and stopping short of the dura to avoid damage. A thin strip of bone covering the superior sagittal sinus was thinned but not removed. Before carefully removing the bone, it was soaked to soften for 5 min with sterile CB. Any superficial bleeding was controlled with small pieces of sterile surgifoam (Ethicon, Cincinnati, OH, USA), previously soaked in sterile CB. The dura was carefully removed by grasping, stretching, and gently piercing it with a 30 G needle, then peeling it off with serrated forceps. The cover glass was carefully placed and initially attached with Vetbond while gently applying downward pressure, ensuring that any gaps between the bone and the glass were filled with Vetbond without reaching the brain. After polymerization of the Vetbond, a few drops of cyanoacrylate glue were added to secure the glass in place, and finally a thin layer of dental cement around the edge of the glass was applied to seal it in place. Typically, this preparation allowed for >6 weeks of cellular-resolution 2p-imaging and >10 weeks of widefield 1-p imaging. If redness or signs of inflammation was seen, dexamethasone (0.5 mg/kg) was given for 3 days.

## In vivo two-photon and 1-p widefield imaging

Mice were typically imaged every 4–6 days, starting between P45 and P55, and every 1–3 days once spontaneous seizures were detected until the animal appeared moribund and was euthanized with $CO_2$. (Average time span between the first and last imaging session was

48 ± 31 (SD) days). The mouse was headfixed and allowed to freely run on top of a custom-built Styrofoam wheel (12 cm diameter), whose rotational speed was recorded by an angular velocity encoder. Both imaging modalities are integrated in a Prairie Ultima IV (Bruker, Billerica, MA, USA) modified in vivo two-photon microscope. One-photon image sequences were acquired with a pco.Edge 4.2 sCMOS camera (pco, Germany), mounted on top of the scan head of the Bruker Ultima IV microscope with a 2x objective lens (CFI Plan Apo Lambda 2X, Nikon, Melville, USA), yielding a raw (unbinned) resolution of 3.62 µm/pixel (FOV = 2060 × 2048 pixels). Widefield movies were acquired using 2x or 4x software binning at 100 Hz (10 ms exposure duration). Blue illumination light (for GCaMP indicators or GFP expressed in tumors) was generated by a Xenon arc lamp (Zeiss, Germany), guided through a 480/20 excitation filter. For tagBFP labeled tumors, a 400/40 excitation and 480/40 emission filter was used. For GCaMP emission and jrGeco1a or RFP excitation, a 525/70 filter was used, and for jrGeco1a and RFP emission, a 620/60 filter was used. For iGlusnfr (Venus) imaging, a 517/20 excitation and 554/23 emission filter set was used. Images were acquired at 100 Hz with 2 × 2 or 4 × 4 digital binning (1020 × 1024 or 500 × 512-pixel FOV, respectively) on a Lenovo P920 workstation (Lenovo, Morrisville, USA). EEG signals were amplified (HP 0.1 Hz, LP 5 kHz, gain x100) using a Model 1700 differential AC amplifier (A-M systems, Sequim, USA), digitized with a USB-6211 multifunction I/O device (National Instruments, Austin, USA) and recorded using WinEDR v3.8.7 freeware (Strathclyde University, United Kingdom) at 10 kHz. The pupil size of the right eye was detected with IR LED illumination and recorded at 30 Hz with a GC660 IR camera (Allied Vision Technologies, Newburyport, USA) using custom Matlab routines (v 2020a, 2021a, Mathworks, Natick, USA). The position and movement of the animal's head, whiskers, front paws, and frontal parts of the body was monitored and recorded at 30 Hz with another IR camera triggered by the WinEDR output signal generator (model SC1280G12N, Thorlabs, Austin, USA). These recordings were used to detect and exclude periods containing movement artifact that might contaminate the pixel by pixel analysis of imaging data.

Two-photon images of calcium and glutamate-reporter activity were acquired with a Prairie Ultima IV 2-photon microscope (software: Prairieview v5.63) using a ×25 objective, 1.1 NA, or a ×16 objective, 0.8 NA, at 920 nm (GCaMP6/7/8) or 1000 nm (jrGeco1a) under spiral (10−20 Hz frame rate) or resonant scan mode (30−35 Hz). We used a 525/70 nm emission filter for GCaMP6/7/8 indicators, a 554/23 nm filter for iGlusnfr, and a 620/60 nm filter for jrGeco1a emission. We targeted primarily L2/3 (mean depth below pia: 163 µm, range 100−240 µm). We also imaged some FOVs in L4 (mean depth below pia: 383 µm, range 360−395 µm), and several FOVs in L5 (mean depth below pia: 570 µm, range 510−640 µm). Laser output power under the objective was kept below 50 mW, corresponding to ~20% of power levels shown to induce lasting histological damage in awake mice[58]. Mice were imaged while awake, head-posted in a holding frame, and allowed to run freely on a circular treadmill.

**One-photon image preprocessing**
Widefield images were processed using a custom-written Matlab pipeline partially based on previously published analysis methods [suite2p/caiman/seqnmf/mesoscale brain explorer/ofamm toolboxes[59−61]]. Briefly, recordings were saved by the pco acquisition software (pco camware v4.13) as multiple 2GB tiff stacks. For the analysis presented in Fig. 5, they were loaded into MATLAB, downsampled to a spatial resolution of ~36 µm/pix, motion-corrected ([normcorre-function], Matlab[62]), multiplied by a mask tiff file to suppress any artefactual signals from the vasculature (manually constructed in imageJ), and converted into ΔF/F movies saved as a single hdf5 file. The other acquired voltage traces corresponding to wheel motion, visual stimulus, IR camera trigger, and EEG were then downsampled and aligned to the ΔF/F traces.

Snapshots (average of 20 frames) of baseline calcium, glutamate, and tumor fluorescence during quiet wakefulness were taken at full resolution (2048 × 2048 pix) and 100 ms exposure time (Fig. 3: tumor fluorescence only, Fig. 4: calcium/glutamate only). To resolve the magnitude of progressive change in neuronal activity and glutamate levels, as well as tumor infiltration rates during these intervals, we quantified the evolution of baseline fluorescence intensity in mice at a repeated sampling time interval ranging from 3−7 days over a period of 6−11 weeks (Fig.3: $n = 11$ 3xCR mice, $n = 8$ GPC6 mice; Fig. 4: $n = 10$ 3xCR mice with calcium indicator, $n = 5$ 3xCR mice with glutamate indicator, $n = 8$ GPC6 mice with calcium indicator), as determined by survival. For quantification of the changes in tumor fluorescence, we show both the CV/d (Fig. 3C) and expansion rate (Fig. 3D). The latter is calculated by simply dividing the area covered in tumor signal by the number of days between recordings. To characterize the local tumor expansion at the surface, the CV/d (calculated for each pixel and then averaged) measures changes in tumor coverage independent of whether these are caused by cell proliferation, cell growth/change in morphology, and cell movement. Therefore, it also captures motility and infiltration dynamics. We calculated the coefficient of variation (CV) for each pixel between consecutive imaging time points for calcium, glutamate, and tumor indicators by first scaling all images from each animal equally, then calculating the ratio of images taken at consecutive recordings, calculating the standard deviation across of all pixels, dividing by the number of days between the sessions, and normalizing by the mean intensity of all pixels of the resulting ratio image (see Figs. 3, 4). The colorized panels in Fig. 3 show a relative increase in tumor infiltration (CV/day) in red (up to +50%) and a decrease, i.e., movement of tumor cells (putatively away from where they were previously located in x- y- or z-direction) in blue (up to −50%, white = no change). Note that these colors do not necessarily represent tumor growth, but any change in tumor location, spread, or size. To summarize the overall progression, we binned the recordings into time intervals of 10 days between P45 and P135 and computed the 90th percentile of the resulting distributions for each bin to account for potential sampling bias towards more frequently sampled animals and potential outliers. Glutamate and calcium snapshots were processed in the same way, and the colorized CV/day panels in Fig. 4 represent dynamic changes in calcium and glutamate accumulation between imaging time points.

**Analysis of widefield calcium signal in relation to local tumor expansion**
To correlate spontaneous calcium activity patterns recorded over time with changes in tumor spread (Fig. 5), we used mesoscopic one-photon calcium imaging of bilateral cortical FOV's spanning on average ~45 mm² at 100 Hz frame rate in five 3xCR and five GPC6 tumor animals. Tumor coverage of the FOV was measured for each recording, and tumor expansion rates were calculated in µm²/day. Based on our observation that tumors generally either existed in a prolonged state of slow expansion (<$10^5$ µm²/day) or entered an accelerated rate of >$10^5$ µm²/day (Fig. 3D), we divided the neural recordings into slow expansion or fast expansion categories. Detrended and normalized ΔF/F image sequences were spatially downsampled by a factor of 8, resulting in a final resolution of ~240 microns per pixel. Active running and whisking episodes were computed and excluded to select images corresponding to 300 s of quiet wakefulness for each recording (supplementary Fig. 1A): Briefly, the voltage signals from the wheel rotary encoder were recorded, downsampled to match the image frame rate and aligned with the images. Using custom routines and calibrations, this signal was converted into absolute speed (not discriminating between forward and backward motion). It was thresholded at 0.043 m/s to ignore brief episodes of rocking or grooming (grooming was recorded as part of the whisking routine). Episodes of

active whisking were identified by loading the avi-file recorded by the wide-angle lens IR camera into Imagej, cropping a segment of the video which only included whiskers against a dark background, and saving this section of the video as a tiff stack. The stack was loaded into Matlab, and the variance of each pixel across all recorded frames was calculated. Variances were averaged and thresholded at an individual level for each recording that maximized sensitivity and specificity toward extracting significant periods of whisking. For each pixel of the mesoscopic cortical fluorescence image sequences, a ΔF/F trace was extracted, and calcium events were identified using a thresholding algorithm, taking into account variations in the baseline noise level for each pixel. From here, we defined the following metrics that provide information about different aspects of the calcium activity patterns observed (supplementary Fig. 1C): (1) "activity per min": the mean of the ΔF/F trace computed over the 300 s trace, (2) "events per sec": the rate of identified calcium transients after thresholding at three SD above the median, divided by 300 s, (3) "mean amplitude": the maximum ΔF/F value of each calcium event averaged over all identified events per pixel. This corresponds to a measure of the maximal firing rate achieved during a single calcium transient. (4) "mean duration": The duration of each event transient was averaged over the 300 s trace for each pixel. This corresponds to a relative measure of sustained firing rate during a calcium transient. Changes in these metrics for all FOV pixels were averaged for each recording, and for each animal, recordings during slow and fast tumor expansion phases were pooled together. These groups were then compared across tumor genotypes using a nonparametric t-test equivalent (Wilcoxon rank-sum test for comparisons of two groups, or Kruskal–Wallis test with Bonferroni multiple comparison correction for comparisons between more than two groups ("KW/mc").

For each recording session, a reference image of the tumor (using either a green/red filter for RFP, blue/green for GFP, or 400 nm/blue for BFP) was acquired, spatially downsampled, and aligned with the calcium signal image to obtain a pixel mask of the tumor mass location. The local tumor expansion rate for each period between recording sessions was calculated by dividing the difference in tumor fluorescence between consecutive recordings by the number of days in between. Tumor spatial coordinates were extracted, and the distance between each pixel and the nearest part of the tumor edge was computed (Supplementary Fig. 1b). For the analysis shown in Fig. 5D, we pooled pixels inside the distance band closest to the tumor (<0.75 mm), and separately all pixels in all distance bands further than 3 mm from the tumor. For each recording in all 3xCR and GPC6 animals, we compared the near and far distributions by dividing their means, generating one ratio value per recording. We then separated those ratios into groups of "3xCR during fast expansion", "3xCR during slow expansion", "GPC6 during slow expansion", and "GPC6 during fast expansion" for the metrics "mean ΔF/min", "amplitude", "events/sec", and "event duration". A nonparametric Kruskal–Wallis test with multiple comparisons correction was used to ascertain statistical significance. Additionally, a linear regression between all tumor distances and the five metric values was performed and adjusted $R^2$ as a goodness-of-fit parameter calculated. The tumor distances were circularly shuffled 500,000 times, and the regression was performed for these bootstrapped repetitions (Supplementary Fig. 1d). Significance (p value) was then computed as the fraction of bootstrapped trials with a higher $R^2$ than the actual value. Significance was quantified as a function of tumor expansion rate. For recordings with significant changes, FOV pixels were then divided into distance bands at 0.75-mm steps from the tumor, and distance-related changes were depicted as shaded error bar plots showing the defining features of the metric distributions in each band (Supplementary Fig. 8).

## Analysis of two-photon cellular calcium signals and network activity

Raw tiff image stacks were loaded into Matlab, motion corrected using normcore, and CNMF was applied to identify active neurons in the FOV. ΔF/F traces were computed for each neuron, followed by deconvolution and neuropil contamination correction (adapted from suite2p-wrapperdeconv function[59] using an algorithm based on previously published work[63], yielding a deconvolved ΔF/F matrix (termed "dΔF/F") that was used for all subsequent analysis steps. This represents an estimate of the current probability that one or multiple spikes were produced within a time bin corresponding to the inverse of the scan rate (typically ~10–12 Hz for spiral scan and 15–30 Hz for resonant scan). Therefore, bursts of spikes occurring within ~33–100 ms generally show up as one distinct "event", as do single spikes. Next, whisking, wheel velocity, EEG, and photodiode voltage traces were downsampled and aligned to the dΔF/F traces. Periods of active whisking and running were excluded for the remainder of the analysis. Deconvolved traces were then thresholded by identifying the bottom 20th percentile of all data (corresponding to random noise), and its median and SD. After thresholding above median + 3 SD, individual deconvolved calcium transients were identified, and four metrics were extracted: (1) overall (averaged) dΔF/F per min, (2) events/sec, and (3) amplitude of the transients. The fourth metric we computed for each neuron was the clustering coefficient (CC): To identify coactive clusters of neurons comprised of >2 neurons, the clusterONE algorithm (BrainConnectivityToolbox, matlab[64,65]) uses the previously computed pair-wise Pearson correlation coefficient matrix as its weighted, undirected network input, and identifies potentially overlapping groups of at least 3 associated neurons. Clustered neuron ensembles have previously been identified from neuronal calcium activity data using very similar approaches[35–37]. The CC of each neuron represents the tendency of its neighbors to be interconnected (see equation (9) in ref. 66). See supplementary Fig. 2 (supplement to Fig. 1) for visualization of example neurons and computation of the four metrics.

We acquired these neuronal activity measures from FOV's at different locations (inside and outside the tumor margin), at different time points, and across tumor genotypes (3xCR, GPC6, and non-tumor control). To analyze chronic changes in activity patterns, we split up the GPC6 and 3xCR recordings into three time bins: early (P41–49, mean 43.6 ± 0.84 sem), mid (P50–56, mean 54.9 ± 1 sem), and late (P70-99, mean 89.2 ± 4.3 sem) in order to compare across 3 dimensions: time, distance from the tumor, and tumor genotype. The control late group spanned P80-P129. For the comparison between areas inside the tumor margin and tumor cell-free surrounding areas, we included data only from mid to late time bins, as we could not ensure that earlier time points left enough time to uniformly express the calcium indicator in all AAV-injected animals. For each animal, we chose FOV's that were either located within the infiltrating tumor margin, i.e., neurons were visibly intermixed with tumor cells (0–1 mm from the solid (neuron-free) tumor core), or located within a 'leading edge' band of 1–2 mm from the tumor margin, where there were no tumor cells visible inside the FOV. We verified the lack of tumor cells in these leading edge FOVs by acquiring z-stacks, but could not rule out the presence of tumor cells beyond a depth of >600 μm due to imaging limitations. Therefore, analogous statistical comparisons were performed independently across these three dimensions: First, we randomly chose equal numbers of neurons from each recording to ensure equal weighting of animals in each group, based on the FOV in each group with the lowest number of identified neurons (3xCR early: $n = 96$, four animals, 3xCR mid: $n = 130$, five animals, 3xCR late: $n = 140$, four animals (all outside the margin); 3xCR inside: $n = 40$, four animals; GPC6 early: $n = 52$, five animals, GPC6 mid: $n = 105$, five animals, GPC6 late: $n = 65$, four animals (all outside the margin); GPC6 inside: $n = 60$, four animals; control mid: $n = 100$, four animals, control late: $n = 100$, five animals). Using 2p-imaging we studied a total of seven 3xCR animals, six GPC6 animals, and eight control

animals, and unique datasets from some animals were included in several groups (see Supplementary Table 1). For each comparison between groups, pooled data distributions, which were generally not normally distributed (see violin plots in supplementary Fig. 9), were generated by randomly choosing 150 neurons from the equally pooled groups for Fig. 6 and 200 neurons from the groups for Fig. 7, and compared using the nonparametric Kruskal–Wallis test, and for each group the significance threshold was set at α = 0.05 with Bonferroni correction to account for multiple comparisons (termed "KW/mc test"). In supplementary Fig. 9, only the percent change between compared groups was displayed with bar plots if deemed significant. In Figs. 6, 7, all bar plots with individual data points) generated from pooled data were plotted, but only meaningful group comparisons, as schematically outlined, were highlighted when significant.

## Statistics and reproducibility

To ensure results were reproducible, data from multiple independent IUE cohorts of 3xCR animals (two cohorts for two-photon imaging, three cohorts for widefield imaging), and GPC6 tumor animals (two cohorts for all imaging data) were included for statistical comparisons. Animals with the same tumor genotype showed similar results regardless of which cohort they originated from. The detailed table in the supplementary material lists the number of animals that were used to replicate results for each figure. Unless stated otherwise for specific experiments, data groups were compared using nonparametric Wilcoxon rank-sum tests for comparisons between two groups, and Kruskal–Wallis test with Bonferroni multiple comparisons correction for comparisons between more than two groups. Sample sizes were chosen based on previous studies analyzing seizure and EEG data using the same mouse tumor models[8,19]. Data from some mice were not included if these animals died before the relevant time points for analysis were reached, or if image quality due to cranial window overgrowth did not permit identification of an appropriate number of neurons and/or appropriate SNR (for 2p-imaging data). Due to the serial chronic nature of the imaging studies following tumor evolution, experiments were not randomized.

## Reporting summary

Further information on research design is available in the Nature Portfolio Reporting Summary linked to this article.

## Data availability

The RNAseq data (Fig. 2) is publicly available at https://www.ncbi.nlm.nih.gov/geo/query/acc.cgi?acc=GSE263832. Data for Suppl. Fig. 4: Single-cell RNA-Seq analysis of adult and pediatric GBM from the publicly available dataset (REF: Neftel et al. Cell 2019, PMID: 31327527); spatial transcriptomic data from the publically available dataset (REF: Puchalski et al. Science 2018, PMID: 29748285); Data for Suppl. Fig. 5: generated from the GlioVis tool (REF: Bowman et al. Neuro-Oncol 2017, PMID: 28031383). The raw in vivo mouse imaging data that support the findings of this study are available under restricted access for reasons of sensitivity, access can be obtained by contacting the lead author (jfmeyer@bcm.edu) within 2 weeks upon request. Data are located in controlled access data storage at Baylor College of Medicine. Source data of all graphs in figures are provided with this paper.

## Code availability

All custom Matlab code used to analyze data supporting the conclusions in this study is available on github.com via Zenodo: https://doi.org/10.5281/zenodo.10954324.

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

## Acknowledgements

The authors wish to thank Ryan Ostrom and Juan Enrique Villacres Perez for their technical assistance. Supported by NCI R01CA223388 (J.N. and B.D.), R01NS124093 (B.D.), R50CA252125 (K.Y.), R01CA263628 (J.M.), and Blue Bird Circle Foundation (J.N.).

## Author contributions

Conceptualization and writing—review and editing, J.N. and B.D.; Methodology, writing—original draft, and visualization, J.M. and K.Y.; Investigation, J.M., K.Y., and E.L.-F.; Funding acquisition, J.N., B.D., J.M., and K.Y.

## Competing interests

The authors declare no competing interests.
