## [Peer Review File · Nature Communications]

REVIEWER COMMENTS

Reviewer #1 (Remarks to the Author):

The article by Meyer et al. offers a profound exploration into the intricate relationship between neuronal hyperactivity and glioblastoma growth. Utilizing two syngeneic mouse models, the researchers delve deep into the mechanisms of how neuronal activity is influenced during the progression of this aggressive brain tumor using advanced imaging intravital imaging technologies.

The authors investigate the dysregulation of functional metrics in neural ensembles during tumor invasion. Intriguingly, these changes are not uniform but vary depending on the stage of malignant progression and the proximity of tumor cells. This suggests a dynamic interplay between the tumor's growth and the surrounding neural environment.

They show that the significant elevation in neural activity occurs during periods of rapid tumor growth. This heightened activity could have profound implications for glioblastoma patient's neurological function and overall well-being. The study further uncovers that abnormal glutamate accumulation, a neurotransmitter involved in neural excitability, precedes and even exceeds the spatial extent of baseline neuronal calcium signaling. This uncoupling in the tumor-infiltrated cortex indicates a disruption in the normal neural signaling processes, which could be a potential therapeutic target. These findings not only enhance our understanding of the disease but also underscore the potential for precision genetic diagnosis. By identifying these specific patterns, one might be better equipped to tailor treatments for glioblastoma patients.

While the main findings of the article are interesting several details of the methodology and results would benefit from a clearer presentation.

I have the following recommendations for improvement:

1. How frequently do EEG seizure episodes align with calcium signaling events? Are the authors able to deduce anything regarding the prognostic capability of calcium imaging? How often is there a mismatch? Ultimately, how reliable is calcium imaging as a method?
2. Why exactly was GPC6 chosen ? Can the authors make this more clear in their results section and compare their findings to findings from other glypicans ?
3. How well and reproducibly infect the AAV which portion of neurons ? The authors write simply "AAV-GCaMP, jrGeco1a, and iGluSnFR". This leaves open which cell type was infected and whether e.g. a synapsin or CamKIIA promoter was used. The authors need to show with e.g. NeuN stainings what they exactly sample from with their AAV strategy.
4. "Figure 1D: Long-term optical stability of cranial windows: widefield (top) and 2-p (bottom) images from the same animal at P62, P127, and P189. Scale (top) = 1 mm, scale (bottom) = 0.1 mm" Are these images from a tumor-bearing mice ? If so please also show how the tumor develops.

5. Figure 2A: From the results section as well as the methods section it is not clear to me which publicly available datasets were used. I assume that single-cell RNA-sequencing data were not used for this purpose. If not I would recommend doing so and also would be interested to see whether there might be an enrichment in certain cell states for GPC6. Lastly, I would clearly separate IDH-mutant astrocytoma and oligodendroglioma as well as adult glioblastoma and pediatric high-grade glioma. For all these disease entities, publicly available data are available (e.g. Neftel et al., Cell 2019, Venteicher et al., Science 2017, Tirosh et al., Nature 2016).
6. Figure 2C: Can the authors estimate whether GPC6 is really expressed in tumor cells or could this also be due to microenvironmental cells expressing GPC6 ? Could the authors also analyze this with e.g. scRNAseq data in tumor cells and compare it to the microenvironmental cells that were sequenced in the same batch ?
7. I have troubles understanding Figure 2G/H and the associated interpretations. First, how do the authors define tumor margin and can this be done uniformly across their models ?
8. Could this be due to a bias in the brain regions that were sampled from ? E.g. that GPC6 tumors were more associated with cortical regions as compared to 3xCr animals ?
9. The authors interpret their results as following: “found a significant increase in excitatory synapses but no significant change in inhibitory synapses in the presence of GPC6.” However, what they apparently analyze is the colocalization of pre-and postsynaptic markers. Why do they analyze the colocalization ? Would the authors not expect a 100 % colocalization in a technically ideal scenario, irrespective of whether you have more or less synapses ?
10. How do they analyze the colocalization as I cannot find a respective methods section to this topic ?
11. I would recommend to quantify the synapse densities instead of a mere colocalization.
12. Figure 2G/H: How do the authors define tumor margin ? Please explain this also for all other findings where they talk about tumor margins.
13. Were the antibodies they used validated with positive and negative controls ? If so, how and please include the data. The PSD95 signal sometimes shows clusters around ~300 nm and I would argue that this could very well come from unspecific labelling as this is quite near the diffraction limit of light.
14. Why is the intensity of vGluT1 so different in their representative sections of both animal models ?
15. The authors use CV/day (i. e. coefficient of variation of pixel-wise daily rate of intensity fluctuation). Looking at the images I am wondering why this measurement was chosen, how well it really captures tumor volume changes (at least via my visual inspection I am not sure whether it really shows tumor growth that I would also see. How does this approach fare in comparison to manual segmentation and/or machine learning-based pixel classification. The authors need to better explain their reasoning here.
16. Are the authors also able to quantify neuronal degeneration in vivo using e.g. their averaged calcium or glutamate indicator signal ? If so could the authors also include this longitudinal analysis ?

Minor Points:

1. The authors write “. At the leading edge, a complex sequence of molecular and cellular remodeling leading to a hypersynaptic microenvironment 4,5 with early loss and impairment of interneurons 6–8 promotes epileptogenesis and impairs cognitive processes.”- I am not sure what a “hypersynaptic” microenvironment exactly means. I would recommend rephrasing this part.
2. The quality of certain figures makes it hard to understand what has been done. A higher resolution of these figures is needed (e.g. 1A)

Reviewer #2 (Remarks to the Author):

Comments to the authors:

The authors used cellular and widefield imaging of calcium and glutamate reporters in two GBM mouse models to assess how GPC6 expression alters neuronal network dynamics. Their key findings include that measures of neuronal activity are dysregulated at a sub-millimeter scale and in a tumor-type-specific manner. Additionally, they show that glutamate and neuronal calcium activity are decoupled in the peritumoral environment. Overall, I find their study's premise and the results compelling and of great interest to those interested in cancer neuroscience. It adds to a small but growing body of work focused on the importance of tumor-induced neuronal alterations. However, the work presented forces the reader/reviewer to make several assumptions, and thus, I have some concerns that will likely need to be addressed before publication of the final version of this manuscript.

Major Comments:

Section: Tumors with and without GPC6 generate distinct cortical infiltration dynamics.

1. In Figure 1E – though it is not the point of the figure – there is an extensive hemispheric tumor signal in the lower panel compared to the upper panel. Are these two mice from the same model? If so, it would suggest a great deal of variability of the growth rate and spatial growth patterns, and thus have implications for how one interprets the study. Representative histological sections of the model at these/similar time points would also be appreciated if available and serve as a complementary figure to that shown in 1E.
2. The Kaplan-Meier curve in Figure 2B consists of both low- and high-grade human glioma cases. Given the very different biology between these two types of glioma and the possible variable influence of GPC6 in the setting of either – this KM data should be presented for LGG/IDH mutant tumors high/low GPC6 and GBM/IDH wildtype tumors (high/low) GPC6.

3. Does the expression of GPC6 vary spatially in human glioma? Are there differences in GPC6 in the core of the tumor vs. the periphery/infiltrative margins where there are more neurons and GPC6 expression is potentially more meaningful?

4. No evidence of GPC6 expression in the tumor model is presented. Immunocytochemical or protein-based/Western blots or transcriptional studies demonstrating the presence/absence of GPC6 expression should be presented to confirm/illustrate the phenotypic differences in the model.

5. Ideally, the bulk RNA sequencing and IF studies investigating synaptic changes in the tumor microenvironment would come from the same time. Is there a reason why this isn't the case? This is of particular interest as the GO results suggest some changes in the GO results suggest some changes in GABAergic signaling – with the following terms being selected [GABA-receptor activity, GABA-gated chloride ion channel activity, chloride channel activity], whereas the IF studies done at P30 suggest no changes in the number inhibitory synapses. If possible/within the scope of the study, data from both the early and late time points would be of interest and would be complimentary to some of the 2P results presented later.

6. What are the DAPI-positive cells in 2G and 2H (middle image) – neurons, tumor cells? It would be useful to see single images of all assessed channels/antibodies in addition to the merged ones.

7. Perhaps my interpretation of the author's results in 2G and 2H is slightly different than what they report. They report that the GPC6 model has a significant increase in excitatory synapses but no differences in inhibitory synapses. This is based on immunohistochemical analysis of the colocalization of PSD95/VGLUT1 and Gephyrin/VGAT. PSD95 and Gephyrin are glutamatergic, and GABAergic post-synaptic scaffolding proteins, and VGLUT1 and VGAT are glutamatergic and GABAergic markers. Given this, I believe what the authors have shown is that in the GPC6 model, the number of glutamatergic synapses on glutamatergic cells has increased, while the number of GABAergic synapses on GABAergic cells has not changed at that particular time point (P30). Though the difference is subtle, I think it is of great importance in the context of the study.

8. The authors use serial mesoscopic measures of fluorescence in GFP-labelled tumor cells and 2-photon imaging to make the claim that there are differences in the cortical infiltration pattern over time of these two models. Again, I think this claim would be better supported by representative immunohistochemical or immunofluorescence images from mice at the studied time points. Additionally, the authors make the claim that the GPC6 tumors first infiltrate aggressively but grow less dramatically once they reach cortical layers 1-4. I don't understand the claim – are the authors suggesting that the initial subcortical infiltration is aggressive? This is based on the FOV coverage per day. What is the basis for this layer-specific claim?

Section: Cortical excitability and glutamate dynamics are uncoupled during GBM progression and depend on tumor genotype

9. Figure 4: An aligned reference widefield fluorescence image of the tumor/its fluorescent marker should be displayed for each time point to put the observed calcium and glutamate signals into the correct spatial context. Without this it is difficult to interpret the spatial relevance of the calcium and glutamate signals relative to the location of the tumor.

10. Along similar lines, if possible, did the authors consider using a distance metric - similar to what was performed in figure 5D - to correlate the intensity of the calcium and glutamate signals relative to tumor location and see how this varies with tumor progression?

11. IHC/IF images of the injected (thy1-gcamp6s, AAV-FLEX-iGluSnFR + AAV-Ef1alpha-Cre) animals should be presented in the figure or in supplementary images with concomitant markers for neurons and tumor cells.

Section: Neural activity patterns are significantly elevated during periods of accelerated tumor growth

12. The authors binned tumor growth rates into high and low growth rates and then assessed neural activity within <7.5 mm of the tumor edge. It is not clear to me from the figure or the manuscript body/methods section what GECI is used in this particular tumor model. Was the syn-driven jRCaMP1a AAV used or the combination of FLEX-GCaMP8m and CamKIIa-Cre. This has implications for how the results in the rest of this section are interpreted.

13. I can't make out the lower portion of the figure in Figures 5A and 5B or the axes in 5F and 5G. The resolution and magnification currently available do not permit reviewer assessment.

14. The authors report that greater neuronal activity was observed during periods of fast tumor growth in both models – as evidenced by increased mean activity levels and greater amplitudes of the observed calcium events. Again, it is unclear if this is pan neuronal activity or only excitatory CamKIIa neurons.

Section: Tumor neuronal crosstalk depends on GPC6, tumor distance and growth rate.

15. This section is somewhat unclear – particularly the subsection which seeks to explain Figure 5F. This is compounded by the small size and poor resolution of the figures available to the reviewer.

Section: GPC6-dependent changes in cellular activity patterns inside versus outside the tumor margin.

16. As in earlier sections – it is unclear which AAV model was used, and thus it is unclear if the neuronal activity being tracked is pan-neuronal or specific to CamKIIa neurons only.

17. Again, the resolution and magnification of the images presented make it difficult to interpret 6A and the images in 6D.

General:

18. Perhaps it is beyond the scope of this work, but it is not clear how you link the findings of decoupled glutamate and neuronal calcium levels to the results observed with 2P and mesoscopic imaging studies.

Reviewer #3 (Remarks to the Author):

Review of Meyer et al – Nature Communications, 2023

In this study “Glioblastoma disrupts cortical network activity at multiple spatial and temporal scales”, by Meyer et al., the authors explore how neuronal activity and glutamate signaling is altered in 2 different models of GBM. GBM is a devastating disorder, has unique progressions based on tumor genetics, and desperately needs novel treatments. This study comes from an accomplished group with expertise in cancer, neuroscience, epilepsy, and in vivo imaging. The question is of high importance and studies are performed rigorously. The central findings presented are that the two GBM variants used (3xCR and GPC6) have different growth rates over time, invade different areas of cortex, and are correlated with unique changes in glutamate and neuronal activity. The authors highlight that there is an interesting and unexpected de-correlation between glutamate and Ca in the 3xCR mice, that there is a clear relationship between distance to the tumor and neuronal activity in the 3xCR mice but not the GPC6 mice, and that the timing of changes in neuronal activity are unique in each model. This is a very novel and cutting edge study and, as mentioned above, comes from experts in the field and is performed extremely rigorously. The central issue with the study, however, is the difficulty in understanding what imaging was done, how it was analyzed, and what it means. The authors try to convey their complex findings to the reader but it is often extremely difficult to understand what the data is (missing figure legends, challenging text, no axes labeled) and how to interpret it (so many different assays, not sure what each means and if it is important). I applaud the authors on their models, questions, and imaging, but the presentation of the data needs improvement so that the reader can understand what they are reading. Overall, I remain highly enthusiastic for an exciting and rigorous study that if presented properly could be very impactful.

Major issues:

- In general, both the figures and the text are quite difficult to follow. The figures tend to lack labels, fonts are very small, and images are hard to see. The text is filled with jargon and the meaning of the data and analysis is often lost due to the dense and technical writing. The study's impact would be significantly increased by improving the way the message is conveyed to the reader. In addition, the text changes between past and present tense which is difficult to read.

- A better explanation of what CV/day is and how it was used to analyze data in Fig. 3 is needed. It is not explained in such a way to favorably interpret the data. Why was CV/day used as a measure rather than total tumor area or something more simple? The issue that occurs with CV/day is that when sequential timepoints are considered, if the tumor does not grow, the DCV/day value is negative (blue). This suggests the tumor is shrinking but I don't believe that is the case. It seems like an overly complex and somewhat confusing way to quantify tumor growth.

- Perhaps a large section of the methods, where image analysis approaches are described should be moved to the results section. The analysis the authors use is custom made to deal with their type of data they have so it would be appropriate to describe their approaches in the results section.

- The logic behind the uncorrelated glutamate and Ca signaling in the 3xCr mice is a bit hard to follow. Wouldn't Ca/neuronal activity still be increased if glutamate is increased? The author suggest that this reflect glutamate from a non-synaptic origin, which is very reasonable based on published GBM studies, but the increased glutamate should still drive local neuronal signaling. Also, Ca signals are likely dominated by somatic, not synaptic, Ca changes again underscoring the confusion in this interpretation.

- The interpretation of the Ca/glutamate relationship (uncorrelated in 3xCr mice) is also difficult to integrate with the fact that Ca activity is high near the tumor and decreases as you move away from the tumor border. Perhaps a model/cartoon would help the central points of the study be better conceptualized?

- A model or cartoon depicting the overall changes across models, times, locations and assays (Ca/Glut/EEG) could be extremely useful. There are so many changes presented that it is challenging to synthesize it all. In addition, A small cartoon or laminar picture may help the reader better interpret the laminar data shown in Fig 3E.

- I have read and re-read Fig 4 and I honestly do not understand what is presented. The Glutamate and calcium image, I believe, are averages of periods of quiet restfulness (300 sec) and then DF/F is calculated and the CV is computed from within that data and plotted? Are the B&W fluorescent images DF/F? Then below are the growth images of the tumor, but because they are color scaled and presented in an equally difficult way to follow (please see above) connecting what is shown in the top and bottom of each figure is extremely challenging. Finally, there are no labels on axes in Fig 4E and F again leading to the readers wondering what they are looking at. This is an extremely complex and high quality data set. I have no doubt of the rigor and precision that went into collecting this amazing data. It deserves a better presentation that allows the reader to understand what they are looking at.

- Specific issues with Figure 4: How were Ca and glutamate images generated for Fig 4? Were static images used or were images generated based on some extrapolation from time series data? Static images are much less informative for Ca and glutamate since signals are so dynamic in time. The Y axes need to be labeled in Fig 4 E and F. In Fig 4 – only 2 glutamate imaging GPC6 mice were used? N = 2 is not sufficient for statistical analysis.

- How was the fast/slow growth cut off determined for Fig 4? Based on fig 3, it seems that there is no slow growth period for 3xCr as it grows consistently fast at all stages.

- In figure 5B, GPC6 tumors show no difference in events/sec in fast vs slow growth phases but in Fig 5C there appears to be as significant difference in the same measure? Please clarify. Is everything in Fig 5 Ca imaging? No glutamate?

- How is seizure activity correlated with the different phases of tumor growth, glutamate signaling, and Ca activity? Is there a meaningful relationship between seizure progression and any of the assays? Is the progression of seizure or any seizure phenotype different between the 2 models?

- Why do the authors believe there is such a different relationship between tumor location and neuronal activity in the 2 models? That seems to be one of the central findings.

Minor Issues:

- Line 471-472 – “When we quantify all the measured events in a given frame during fast and slow growing stages in our 3xCr brain (Fig 5A), we observed 13% fewer events/sec.” In which group? Unclear

- Was all imaging data taken from periods of animal stillness? How were the behavioral state quantifications integrated into the analysis?

- What does “Tumor neuronal crosstalk” mean? This is an example of jargony terms that should be avoided.

Reviewer #4 (Remarks to the Author):

Overall

The work by Meyer et al. characterizing the influence of glioblastoma on neuronal changes tackles a set of very important and challenging questions in the field with a set of novel and exciting approaches. The authors focused on the spatial and temporal differences occurring in neurons between two mouse models of glioma, one with and one without an addition of GPC6. The authors found noteworthy results in synapse markers, widefield glutamate and calcium changes, and single neuron event rate changes. There are a number claims and conclusions made in the paper that are not directly supported by the presented evidence, as detailed below. Additionally, there are a number of clarifications that need to be made regarding the data analysis and interpretations of results before publication. The work is exciting and novel and would be a great addition to the literature once properly revised.

Recommendation: Revise

Overall

The work by Meyer et al. characterizing the influence of glioblastoma on neuronal changes tackles a set of very important and challenging questions in the field with a set of novel and exciting approaches. The authors focused on the spatial and temporal differences occurring in neurons between two mouse models of glioma, one with and one without an addition of GPC6. The authors found noteworthy results in synapse markers, widefield glutamate and calcium changes, and single neuron event rate changes. There are a number claims and conclusions made in the paper that are not directly supported by the presented evidence, as detailed below. Additionally, there are a number of clarifications that need to be made regarding the data analysis and interpretations of results before publication. The work is exciting and novel and would be a great addition to the literature once properly revised.

Recommendation: Revise

Major Comments:

General:

- Across the entire paper the y-axes on the plots are not consistently labeled. This needs to be fixed
- There seems to be a large difference between the time points along tumor progression that various analyses are done, which prevents one from understanding how the different analyses can be viewed together. For example, the staining performed in figure 2G-H looking at synaptic changes is done at P30 and the functional analyses are performed P40-P120. This needs to be addressed. If available, they should provide the histological analyses of the late stage tumors. At the very least they should discuss the limitations that this discrepancy puts on any correlation between the histological and functional analyses.
- There is a presumed starting point of this paper that there is a known increase of synapses at the margin of glioma, a "hypersynaptic network". This evidence is not shown in this paper and I know of no convincing evidence in the literature of direct measurements of increased synaptic density at the glioma margin. To provide direct support for this claim, they should provide

comparative analysis to control brains.

Methods:

- The authors need to provide more details on how they performed their 4 “metrics” of analysis for the 2P data set

o Specifically, this figure should address the following issues which are currently unclear:

1. What is meant by “overall (summed) $d\Delta F/F$ ”? What constitutes an “event” and what does this “event” look like?

2. How did the authors determine the “duration” of a calcium transient on deconvolved data? Presumably when deconvolved, the calcium event will be discretized into a discrete number of single spike times.

3. How did the authors account for the slow decay of calcium fluorescence in determining “coactive clusters” ie how was the temporal window determined under which clusters were coactive? Neither of the cited works for the clustering coefficient analysis include an application to calcium imaging data

Figure 2:

- For the data in Fig2G-H, it would be very informative to also plot PSD95, VGlut1, Gephyrin, and VGAT levels alone in these different

Figure 3:

- It appears that the fluorescence tumor signal disappears after P83 in Fig 3B, which could be due to numerous experimental reasons other than tumor growth changes. The conclusion that the data in figure 3 “reveal that 3xCR tumors relentlessly infiltrate cortical territory throughout disease progression, whereas GPC6 tumors first infiltrate aggressively, but grow less dramatically later once they reach cortical layers 1-4” needs to be corroborated by tissue-based analysis of the brains.

- Given the concerning reduction in fluorescence signal across all measurements – tumor fluorescence, calcium, glutamate – in the GPC6 model in the later time points (after ~P60) – it would be very important for the authors to plot all the signals for each animal together and see if they reduce together as time progresses. Perhaps the changes being seen are due to the quality of the cranial window decreasing. This concern could be addressed by correlating the

last time-point of in vivo imaging of tumor burden with the postmortem histological analysis of tumor size.

Figure 4:

- The presentation and explanations of the data in figure 4 is confusing and difficult to interpret.

The data clearly demonstrates dynamic changes in glutamate and calcium but fails to demonstrate any clear patterns or correlations between the two. Interpretations beyond that seem speculative.

- The histologic analysis performed in Figure 2 is done much earlier than the analyses performed in figures 3 and 4, and no quantitative analyses was done to correlate the two. Any conclusions drawn by connecting these two analyses are speculative and should be framed as such. For example, manuscript lines 459-460.

Figure 5:

- The results of figure 5D-G are confusing and need to be clarified more carefully in both the manuscript and figure. For example, in this statement – “In both 3xCr and GPC6 tumor brains, there was a significantly higher event rate proximal to the tumor (p-values .02 and 2e-6, respectively)” – what is the meaning of this comparison to this null distribution? Why not directly compare event rates near and far from the tumor between the two models?

Figure 6:

- There are 4 metrics for each subplot ($\Delta F/F$, events/s, amplitude, clustering) and 3 subplots for a total of 12 comparisons. However, there is only data presented for 4 out of these 12 comparisons. The authors should explain why these other comparisons were not considered.

Figure 7:

- The way the data is organized in figure 7 does not allow the reader to directly compare how each metric ($\Delta F/F$, events/s, amplitude, clustering) changes in the animals across time (early vs. middle vs. late). It would be much easier to understand as each metric simply plotted through time (early, middle, late) with clear indications of which time points have statistical differences. Currently, there are no statistics indicated on the figure.

Minor Comments

- From the low magnification images in Fig 2G-H, it is difficult to orient oneself to where in the

brain the peritumoral analysis has been done. Authors should please provide more clear low mag images to orient where in the brain the tumor and imaging fields are, so that one can properly assess whether the authors have taken brain regions differences into account in this analysis. Additionally, in this figure, the ordering of the groups in the boxplots should match the ordering of the columns of the representative images.

- Authors need to provide more details about how the immunofluorescence images in Fig 2G-H were acquired and analyzed for colocalization.

- The use of “early,” “middle,” “late” timepoint labels is very confusing and inconsistent throughout the paper. On line 352 of the manuscript authors say “(early (P40-49), mid (P-50-59) and late (P60-100)), but later on lines 437 the early is defined as P56 – P69. What time point would the analyses in figure 2 at P30 be considered?

- What do the red and orange colors indicate on the raster plots in Figure 6A and 7A? If amplitude of signal, include color scale info.

General response to the reviewers by the authors:

We sincerely thank all 4 reviewers for the time they dedicated to reading and commenting on our manuscript. All their questions and comments were exceptionally thoughtful and thorough, and as a consequence of addressing them as rigorously as possible, we were able to raise the quality of the manuscript by a significant margin. We made an effort to point out responses to questions that were similarly asked by other reviewers and addressed elsewhere in this document. In the updated manuscript we highlighted all significant changes that were made. We tried to use different colors for changes requested by the individual reviewers (see below this paragraph). We chose to address the questions raised by reviewer 4 first, therefore we refer to those comments when similar questions were asked by reviewers 1-3. Overall, we reanalyzed both widefield and 2-photon data from the majority of animals based on the specific requests concerning tumor- calcium- and glutamate imaging (figures 3-7) and added data from 2 animals (16 individual recordings) that had not been included previously. Furthermore, we added additional supplementary figures, a schematic summary cartoon, and a supplementary table listing all animals used in this paper. We sincerely hope to have resolved the reviewers' concerns and look forward to answering any additional questions that may arise.

Reviewer 1 Reviewer 2 Reviewer 3 Reviewer 4

REVIEWER COMMENTS

Reviewer #1 (Remarks to the Author):

The article by Meyer et al. offers a profound exploration into the intricate relationship between neuronal hyperactivity and glioblastoma growth. Utilizing two syngeneic mouse models, the researchers delve deep into the mechanisms of how neuronal activity is influenced during the progression of this aggressive brain tumor using advanced imaging intravital imaging technologies.

The authors investigate the dysregulation of functional metrics in neural ensembles during tumor invasion. Intriguingly, these changes are not uniform but vary depending on the stage of malignant progression and the proximity of tumor cells. This suggests a dynamic interplay between the tumor's growth and the surrounding neural environment.

They show that the significant elevation in neural activity occurs during periods of rapid tumor growth. This heightened activity could have profound implications for glioblastoma patient's neurological function and overall well-being. The study further uncovers that abnormal glutamate accumulation, a neurotransmitter involved in neural excitability, precedes and even exceeds the spatial extent of baseline neuronal calcium signaling. This uncoupling in the tumor-infiltrated cortex indicates a disruption in the normal neural signaling processes, which could be a potential therapeutic target. These findings not only enhance our understanding of the disease but also underscore the potential for precision genetic diagnosis. By identifying these specific patterns, one might be better equipped to tailor treatments for glioblastoma patients.

While the main findings of the article are interesting several details of the methodology and results would benefit from a clearer presentation.

I have the following recommendations for improvement:

1. How frequently do EEG seizure episodes align with calcium signaling events? Are the authors able to deduce anything regarding the prognostic capability of calcium imaging? How often is there a mismatch? Ultimately, how reliable is calcium imaging as a method?

JM: Thank you for this important question. In the tumor model we used, spontaneous generalized convulsive seizures present in a significant portion of the animals, however they do not occur at a high enough rate for us to be able to record them in significant numbers while imaging, as every animal can only

be imaged for up to 6 hours at a time, every 2-3 days. When we do see them, they are always visible on the EEG as well as in the calcium signal, usually with generalization across the whole cortical surface, but there were just not enough of them to reliably run any statistical analysis. Therefore we did not include any data directly describing seizures or seizure timing in the manuscript. However, implicitly one can deduce that the chronic changes in baseline calcium and glutamate signaling we observed may underlie a reduction of seizure threshold. We have not carried out any specific analysis of preictal time periods with respect to the utility of calcium activity as a seizure predictor. This would certainly be a worthwhile endeavor, but it lies outside the scope of this study. Here we focused on using calcium and glutamate imaging as a measurement to describe how peritumoral neural activity is altered by tumor infiltration over time. In future studies we do anticipate to be able to characterize markers of epileptogenesis, such as changes in excitation-inhibition balance, more precisely. We added language to the discussion reflecting these thoughts.

2. Why exactly was GPC6 chosen? Can the authors make this more clear in their results section and compare their findings to findings from other glypicans?

JM: We agree that this choice is not immediately obvious without any background information. Glypicans were previously shown to be secreted by astrocytes during development to induce functionally intact synapses (as opposed to thrombospondins), we cite [Allen et al, Nature 2012; deWit et al, Neuron 2013; Filmus, Glycobiology 2001] in the manuscript. We had previously shown (Yu et al, Nature 2020) that tumoral overexpression of GPC3 causes reduced survival, increased electrographic seizures and interictal spikes, elevated synaptogenesis, and increased tumor BrDu levels. We decided to use GPC6 in this study, because out of the six members of the glypican family, it had previously been shown to be specifically involved in the promotion of excitatory synapses (Allen et al, Nature 2012). We added a clarifying statement to the introduction and discussion.

3. How well and reproducibly infect the AAV which portion of neurons? The authors write simply “AAV-GCaMP, jrGeco1a, and iGluSNfr”. This leaves open which cell type was infected and whether e.g. a synapsin or CamKIIA promoter was used. The authors need to show with e.g. NeuN stainings what they exactly sample from with their AAV strategy.

JM: Thank you for this observation, we did indeed omit an important disclosure of which neuronal populations exactly were labeled in each dataset, and have added this information to the methods and as a supplementary table. To express calcium indicators, we used one pan-neuronal promoter (h-synapsin), and two different promoters found only in pyramidal neurons (thy1, CamKII α). For iGluSNfr expression, we used one pan-neuronal promoter (EF-1 α), and one pyramidal neuron promoter (CamKII α). These are well-established methods of introducing neuronal fluorescent markers in mice. Expression levels and profiles of calcium and glutamate indicators using these constructs and promoters in murine preparations very similar to our own have been verified to be expressed in the intended target cell population and published before, and we do not think that replicating these studies here would strengthen our results. We have inserted additional clarification and references for the expression of these constructs in the methods section. Additionally, we added a supplementary table with information for all animals used for calcium and glutamate imaging (figures 4-7).

4. “Figure 1D: Long-term optical stability of cranial windows: widefield (top) and 2-p (bottom) images from the same animal at P62, P127, and P189. Scale (top) = 1 mm, scale (bottom) = 0.1 mm” Are these images from a tumor-bearing mice? If so, please also show how the tumor develops.

JM: Thank you for this comment, we actually noticed that there was a mistake in the legend which has been fixed (P189 is not shown in the figure). Moreover, we have added an additional supplementary figure (#6)

with detailed images from 2 representative animals showing calcium indicator and tumor side by side. The point of figure 1D was to show a concise example of the same mesoscopic cranial window over 2 months apart. To demonstrate the consistent optical clarity, we thought it best to show only one channel which lets the reader clearly see the detailed dorsal vasculature. In the new supplementary figure, the reader can get a more detailed impression of what a representative cranial window looks like over time in both the calcium indicator channel and the tumor channel.

5. Figure 2A: From the results section as well as the methods section it is not clear to me which publicly available datasets were used. I assume that single-cell RNA-sequencing data were not used for this purpose. If not I would recommend doing so and also would be interested to see whether there might be an enrichment in certain cell states for GPC6. Lastly, I would clearly separate IDH-mutant astrocytoma and oligodendroglioma as well as adult glioblastoma and pediatric high-grade glioma. For all these disease entities, publicly available data are available (e.g. Neftel et al., Cell 2019, Venteicher et al., Science 2017, Tirosh et al., Nature 2016).

6. Figure 2C: Can the authors estimate whether GPC6 is really expressed in tumor cells or could this also be due to microenvironmental cells expressing GPC6 ? Could the authors also analyze this with e.g. scRNAseq data in tumor cells and compare it to the microenvironmental cells that were sequenced in the same batch ?

KY: We thank the reviewer for these two concerns. The human data analysis was performed through the GEPIA online tool (Tang et al, Nuclei Acids Res 2017, PMID: 28407145) which uses the RNA-Seq datasets from the UCSC Xena project (<http://xena.ucsc.edu>). As this and other comments address, the source of GPC6 is very interesting. To address this, we utilized publically available studies (Broad Institute, https://portals.broadinstitute.org/single_cell/study/SCP393/single-cell-rna-seq-of-adult-and-pediatric-glioblastoma, Neftel et al, Cell 2019). In what is now supplemental figure 4, we can clearly see that the detectable expression of GPC6 is restricted to the tumor cells. Additionally, there are clearly clusters that express GPC6 in both adult and pediatric samples of high grade glioma. To address the differences between subtypes, we performed more detailed survival analysis, isolating different glioma subtypes. This data will now be included in supplemental figure 5.

7. I have troubles understanding Figure 2G/H and the associated interpretations. First, how do the authors define tumor margin and can this be done uniformly across their models ?

8. Could this be due to a bias in the brain regions that were sampled from ? E.g. that GPC6 tumors were more associated with cortical regions as compared to 3xCr animals ?

KY: We apologize for the lack of clarity. Following our previously published approach (Yu et al, Nature 2020, PMID: 31996845; Yu et al, Neuro-Oncol 2023, PMID: 36044040; Curry et al, Neuron 2023, PMID: 36787748; and Huang-Hobbs et al, Nature 2023, PMID: 37380778), we assess the effects of tumors on peri-tumoral neurons. We defined the region as either within 50µm of the leading edge of the tumor or regions within invasive tumors cells. Given our previously reviewed publications, we believe our approach is reliable. While it remains possible that endogenous or non-tumor GPC6 can contribute to peri-tumoral synapses, expression studies through the Allen Brain Atlas and the Mouse ENCODE transcriptome data reveal very poor expression in adult tissues. Therefore, we are confident that the phenotypes we observe are a result of transgenic GPC6 expression.

9. The authors interpret their results as following: “found a significant increase in excitatory synapses but no significant change in inhibitory synapses in the presence of GPC6.” However, what they apparently analyze is the colocalization of pre-and postsynaptic markers. Why do they analyze the colocalization ? Would the authors not expect a 100 % colocalization in a technically ideal scenario, irrespective of whether you have more or less synapses ?

10. How do they analyze the colocalization as I cannot find a respective methods section to this topic ?

11. I would recommend to quantify the synapse densities instead of a mere colocalization.

KY: We thank the reviewers for this comment. The rationale for only quantifying the colocalization of pre- and post-synaptic markers is that functional synapses require the interaction between both. While the presence of only one may suggest a system is better primed for increased synapses, the functional unit requires synaptic communication across cells. Additionally, we follow the published standards of other leaders in the field (PMID: 27615741). Lastly, the quantification presented in the manuscript are a measure of density. They are the count of synapses in a specific frame within our definitions of the tumor margin. We apologize for the ambiguity and have amended the text to provide greater clarity.

12. Figure 2G/H: How do the authors define tumor margin ? Please explain this also for all other findings where they talk about tumor margins.

KY: See response for comment 7 above.

13. Were the antibodies they used validated with positive and negative controls ? If so, how and please include the data. The PSD95 signal sometimes shows clusters around ~300 nm and I would argue that this could very well come from unspecific labelling as this is quite near the diffraction limit of light.

14. Why is the intensity of vGluT1 so different in their representative sections of both animal models ?

KY: We thank the reviewer for this observation. As controls, a section from each brain was subjected to the same staining procedure with the exception that primary antibodies were excluded. This was our secondary only control. We used these sections to set our imaging parameters to exclude any autofluorescence or background noise. We also used these images to set the signal sensitivity in our quantification to ensure no false positive counts. The difference in vGluT1 signal suggests to us that there is an increase in presynaptic markers at the tumor margin. However, as stated above the focus was on the overlap of pre- and post-synaptic markers, which is a requirement for a functional synapse.

15. The authors use CV/day (i. e. coefficient of variation of pixel-wise daily rate of intensity fluctuation). Looking at the images I am wondering why this measurement was chosen, how well it really captures tumor volume changes (at least via my visual inspection I am not sure whether it really shows tumor growth that I would also see. How does this approach fare in comparison to manual segmentation and/or machine learning-based pixel classification. The authors need to better explain their reasoning here.

JM: Thank you for this important observation, we agree that we did not sufficiently clarify why we chose this measurement and have added a statement summarizing the following explanation. For quantification of the changes in tumor fluorescence, we show both the CV/d (fig 3C) and expansion rate (fig 3D). The latter is calculated by simply dividing the area covered in tumor signal by the number of days between recordings, and we clarified throughout the text that we are not claiming to accurately measure tumor growth (see also the responses to a similar questions by reviewers 3 and 4 below), as we can only see the dorsal surface, and the tumors could potentially be growing at different rates subcortically. All we can characterize for certain is local expansion at the surface. The CV/d (calculated for each pixel and then averaged), however, measures changes in tumor coverage more generally, independent of whether these are caused by cell proliferation, cell growth/change in morphology, and cell movement. Therefore, it also captures motility and infiltration dynamics, which is an important aspect of the disease we attempt to study. Ideally, to study infiltration dynamics in isolation, one would follow the same tumor cells over time, but from a technical standpoint this

would be beyond the scope of this work.

16. Are the authors also able to quantify neuronal degeneration in vivo using e.g. their averaged calcium or glutamate indicator signal ? If so could the authors also include this longitudinal analysis ?

JM: Thanks for this question, we agree that this is an important facet of disease progression which could be addressed using our chronic imaging technique. In fact, we do show in figure 3E examples of tumor cells coexisting with calcium-indicator labeled neurons. We have many more of these 3D image stacks from several animals and time points, however we chose not to fully analyze them for this study, because i) calcium indicators are not the best tool for visualizing morphological degeneration, so this would ideally require a new set of animals labeled with a “static” neuronal fluorescent protein such as GFP or tdTomato, ii) We have shown loss of pyramidal and PV-expressing inhibitory neurons, as well as perineuronal nets, in the same 3xCR GBM model we used here (Hatcher et al, JCI 2020), and iii) the changes we show in GPC6 mesoscopic calcium and in cellular-level activity patterns in both 3xCR and GPC6 might be interpreted as a form of “functional degeneration”, but we chose not to explicitly frame the results in that way to avoid confusion with the more common use of degeneration as a description of morphological cell integrity.

Minor Points:

1. The authors write “. At the leading edge, a complex sequence of molecular and cellular remodeling leading to a hypersynaptic microenvironment 4,5 with early loss and impairment of interneurons 6–8 promotes epileptogenesis and impairs cognitive processes.” - I am not sure what a “hypersynaptic” microenvironment exactly means. I would recommend rephrasing this part.

JM: Thank you for this observation. We received a similar comment from reviewer 4 (see our response below) and adjusted the language, pointing out that what the literature appears to show is a “remodeling” of synaptic structures favoring hyperexcitability in the TME.

2. The quality of certain figures makes it hard to understand what has been done. A higher resolution of these figures is needed (e.g. 1A)

JM: We agree that some figures suffered from degraded resolution and quality in the original submitted pdf, and we have made every effort to improve image quality in the resubmitted version.

Reviewer #2 (Remarks to the Author):

Comments to the authors:

The authors used cellular and widefield imaging of calcium and glutamate reporters in two GBM mouse models to assess how GPC6 expression alters neuronal network dynamics. Their key findings include that measures of neuronal activity are dysregulated at a sub-millimeter scale and in a tumor-type-specific manner. Additionally, they show that glutamate and neuronal calcium activity are decoupled in the peritumoral environment. Overall, I find their study's premise and the results compelling and of great interest to those interested in cancer neuroscience. It adds to a small but growing body of work focused on the importance of tumor-induced neuronal alterations. However, the work presented forces the reader/reviewer to make several assumptions, and thus, I have some concerns that will likely need to be addressed before publication of the final version of this manuscript.

Major Comments:

Section: Tumors with and without GPC6 generate distinct cortical infiltration dynamics.

1. In Figure 1E – though it is not the point of the figure – there is an extensive hemispheric tumor signal in the lower panel compared to the upper panel. Are these two mice from the same model? If so, it would suggest a great deal of variability of the growth rate and spatial growth patterns, and thus have implications for how one interprets the study. Representative histological sections of the model at these/similar time points would also be appreciated if available and serve as a complementary figure to that shown in 1E.

JM: Thank you for this observation, we agree that in our IUE tumor preparation there was an appreciable amount of variability regarding tumor growth, which is also reflected in the variability in survival (figure 2D, see also Yu et al, Nature 2020). Directly quantifying tumor growth over time by measuring the total volume (versus what is visible in our cortical imaging) was technically and conceptually outside the scope of this study. We made extensive changes to all sections in the text regarding tumor growth to avoid any confusion. Instead of growth, we are only claiming to analyze the local expansion at the dorsal surface (e.g. plotted in figure 3D), and infiltration/spread dynamics (fig 3A-C). See also the very similar comment from reviewer 4 below.

2. The Kaplan-Meier curve in Figure 2B consists of both low- and high-grade human glioma cases. Given the very different biology between these two types of glioma and the possible variable influence of GPC6 in the setting of either – this KM data should be presented for LGG/IDH mutant tumors high/low GPC6 and GBM/IDH wildtype tumors (high/low) GPC6.

KY: See response for reviewer 1, comments 1-5 above.

3. Does the expression of GPC6 vary spatially in human glioma? Are there differences in GPC6 in the core of the tumor vs. the periphery/infiltrative margins where there are more neurons and GPC6 expression is potentially more meaningful?

KY: This is an excellent point. To address this, we used the IVY-GAP database which provide spatial transcriptomic of glioblastoma tissue. A query into this dataset reveals that GPC6 expression is weakest at the leading edge and infiltrating cells. The most abundant expression is within the tumor. This data is included in supplemental figure 4.

4. No evidence of GPC6 expression in the tumor model is presented. Immunocytochemical or protein-based/Western blots or transcriptional studies demonstrating the presence/absence of GPC6 expression should be presented to confirm/illustrate the phenotypic differences in the model.

KY: We thank the reviewer for this note. System validation is a critical component of any experimental design. The human and mouse coding sequences of GPC6 are 89.33% conserved, suggesting that our mouse RNA-seq analysis should detect the human transcript. A closer inspection of our mouse RNA-Seq data show a significant increase in transcript count presented in the table below. Given this and the strong phenotypic difference, we are confident the differences we observed are from the transgenic GPC6 expression.

Gpc6	216.5075	226.0374	125.6933	675.4478	262.8013	3433.849	5084.445
	3xCR			3xCR; GPC6			

5. Ideally, the bulk RNA sequencing and IF studies investigating synaptic changes in the tumor microenvironment would come from the same time. Is there a reason why this isn't the case? This is of particular interest as the GO results suggest some changes in the GO results suggest some changes in GABAergic signaling – with the following terms being selected [GABA-receptor activity, GABA-gated chloride ion channel activity, chloride channel activity], whereas the IF studies done at P30 suggest no changes in the number inhibitory synapses. If possible/within the scope of the study, data from both the early and late time points would be of interest and would be complimentary to some of the 2P results presented later.

KY: We thank the reviewers for these observations. While there is some discrepancy in the analysis, these experiments provide different details concerning the tumor's activity. The end stage RNA-Seq analysis provides a perspective on the active mechanisms in the tumor brain. However, at end stage, the "horse has left the barn", in that earlier events responsible for tumor progression likely took place months before. Additionally, these end stage brains have massive tumors which will inherently deform and disrupt what remains of the normal brain. This would confound interpretation of the immunohistological analysis.

To test if these are passive downstream effects of the tumor or could potentially contribute to tumor progression, we analyzed earlier time points. Given that these tumors progress at different individual paces based on survival and BrdU studies, we opted to time match rather than growth match the control and experimental samples. While it could be interesting to perform the sequencing at earlier time points, a concordance between age matched sequencing and immunohistology provides minimal additions to the studies' conclusions while requiring substantial resources. Therefore, we respectfully cannot agree with the suggestion to perform these studies.

6. What are the DAPI-positive cells in 2G and 2H (middle image) – neurons, tumor cells? It would be useful to see single images of all assessed channels/antibodies in addition to the merged ones.

KY: We did not specifically identify the non-tumor cells in the peritumoral margin, which are a mixture of neurons, glia, and microglia. We also provide single channel images as requested (figure 2).

7. Perhaps my interpretation of the author's results in 2G and 2H is slightly different than what they report. They report that the GPC6 model has a significant increase in excitatory synapses but no differences in inhibitory synapses. This is based on immunohistochemical analysis of the colocalization of PSD95/VGLUT1 and Gephyrin/VGAT. PSD95 and Gephyrin are glutamatergic, and GABAergic post-synaptic scaffolding proteins, and VGLUT1 and VGAT are glutamatergic and GABAergic markers. Given this, I believe what the authors have shown is that in the GPC6 model, the number of glutamatergic synapses on glutamatergic cells has increased, while the number of GABAergic synapses on GABAergic cells has not changed at that particular time point (P30). Though the difference is subtle, I think it is of great importance in the context of the study.

KY: We thank the reviewer for this consideration. The colocalization of these pre- and postsynaptic markers identify a standard molecular definition of a functional excitatory and inhibitory synapse for purposes of quantification, but do not identify the nature of the postsynaptic neuron. While the latter would be nice to know, given the complexity of cortical synaptic microcircuitry, even with this knowledge, it would not be possible to conclude anything about the overall network behavior. The E/I imbalance is simply a useful starting point in interpreting the altered microenvironment.

8. The authors use serial mesoscopic measures of fluorescence in GFP-labelled tumor cells and 2-photon imaging to make the claim that there are differences in the cortical infiltration pattern over time of these two models. Again, I think this claim would be better supported by representative immunohistochemical or immunofluorescence images from mice at the studied time points. Additionally, the authors make the claim

that the GPC6 tumors first infiltrate aggressively but grow less dramatically once they reach cortical layers 1-4. I don't understand the claim – are the authors suggesting that the initial subcortical infiltration is aggressive? This is based on the FOV coverage per day. What is the basis for this layer-specific claim?

JM: Thank you for this comment. We agree that a dedicated, comprehensive study of tumor infiltration dynamics and patterns would best be performed by IHC/IF images. That way one could theoretically characterize those dynamics around the entire tumor, despite its non-uniform spatial growth however important limitations of such a study would be that i) it would still be a snapshot in time for each animal, so it would be an estimate of infiltration levels (requiring a very large number of animals), and ii) it would still be hard to track individual tumor cells to distinguish between proliferation and spread. In our work, we try to make a distinction between infiltration as captured by changes in tumor fluorescence over time (figure 3C), and local expansion in the dorsal cortex (fig 3D), and we made adjustments to the manuscript to clarify this separation. The point we are making in figure 3D/E is that GPC6 tumors expand aggressively early but then typically slow down after ~P80 (sometimes accelerating again at even later time points), whereas 3xCR tumors do not seem to expand quite as fast at very early time points, but never stop expanding later in the disease. The image in 3E is meant as an illustration of representative cortical patches of 3xCR and GPC6 tumor margin across time showing some differences in cellular infiltration and proliferation patterns between 3xCR and GPC6. We only show this example to demonstrate that there were differences on a cellular level, but did not comprehensively quantify these differences because with our 2-photon FOV size (0.8-0.8mm) it would not have been feasible to sample the entire tumor margin of each animal at each time point in order to appropriately account for spatial peritumoral heterogeneities. We adjusted the text in this section to remove the claim about GPC6 tumors slowing down once they reach layers 1-4, as we agree this was misleading and not appropriately supported by the data we show.

Section: Cortical excitability and glutamate dynamics are uncoupled during GBM progression and depend on tumor genotype

9. Figure 4: An aligned reference widefield fluorescence image of the tumor/its fluorescent marker should be displayed for each time point to put the observed calcium and glutamate signals into the correct spatial context. Without this it is difficult to interpret the spatial relevance of the calcium and glutamate signals relative to the location of the tumor.

JM: Thank you for this suggestion. We made an additional supplementary figure (#7 A: 3xCR calcium indicator, B: 3xCR glutamate indicator, C: GPC6 calcium indicator) with the dual-channel composite images, as we felt figure 4 would have become too busy otherwise.

10. Along similar lines, if possible, did the authors consider using a distance metric - similar to what was performed in figure 5D - to correlate the intensity of the calcium and glutamate signals relative to tumor location and see how this varies with tumor progression?

JM: Thank you for this suggestion. For the particular analysis we performed, the precise question we are trying to answer is: “when a tumor moves in and through the cortical window we image, how do baseline calcium and glutamate dynamics change over time?” (For the spontaneous calcium activity patterns we ask a similar question in figure 5B, but as a function of tumor expansion rate, not time). For the calcium signal, we did run a preliminary analysis of the data shown in 5D as a function of time instead of expansion rate, and found a weaker relationship than the expansion rate-dependent analysis yielded. However, we do show changes in the ratio of calcium activity metrics near versus far from the tumor edge in fig 5C/D. We believe that, in the context of asking whether neuronal populations residing close to the tumor were functionally altered in a different way than those further away, this analysis was more informative than using the calcium baseline images from figure 4. For the glutamate data, this analysis has not been performed, but would certainly be an obvious route to pursue, especially once we have had enough time to collect the

appropriate amount of glutamate images from GPC6 tumor animals.

11. IHC/IF images of the injected (thy1-gcamp6s, AAV-FLEX-iGlufr + AAV-Ef1alpha-Cre) animals should be presented in the figure or in supplementary images with concomitant markers for neurons and tumor cells.

JM: Because we followed each animal during imaging for as long as possible, most of them died naturally as a consequence of terminal disease, and we were not able to harvest their brains for IHC processing. Is the reason for this question that there may be changes to peritumoral neurons that would alter their ability and/or levels of AAV uptake, or that their electrical activity does not translate to calcium dynamics the way it would in WT animals? In that case, we would ideally co-inject these populations with calcium and voltage indicators to test that hypothesis. This would not be an unjustifiable experiment to do, however it is outside our capacity to arrange, calibrate, troubleshoot, repeat and analyze such experiments for this study. Based on the uniform delay between injection and expression level, as in brightness of the calcium/glutamate indicators we observed inside the tumor margins as well as further away from the tumor, we are confident that there is no significant difference in the AAV uptake and intracellular action between intra- and extramarginal neurons. For the two-photon results, we only included animals with calcium reporter expressed in pyramidal cells, because we had previously shown that PV-expressing interneuron health inside the tumor margin is disproportionately degraded, and therefore it would have been problematic to compare all neurons inside and outside the tumor margin with each other. To be safe, we added a statement reflecting this potential caveat to the manuscript. If there is any concern that AAVs may have been taken up by tumor cells, we have and can show (upon request) a large number of two-photon movies recorded in 2-channel mode that, as far as we have observed, never show any calcium or glutamate signal stemming from tumor cells.

Section: Neural activity patterns are significantly elevated during periods of accelerated tumor growth

12. The authors binned tumor growth rates into high and low growth rates and then assessed neural activity within <7.5 mm of the tumor edge. It is not clear to me from the figure or the manuscript body/methods section what GECl is used in this particular tumor model. Was the syn-driven jCAMP7 AAV used or the combination of FLEX-GCaMP8m and CamKIIa-Cre. This has implications for how the results in the rest of this section are interpreted.

JM: We have added a table to delineate which animals were used in which analysis and what their calcium/glutamate indicators were (see supplementary material). We have listed exactly which animals were used for which section of the data analysis by figure. For figure 5B, we wanted to compare animals that had similar life spans and were thus able to be recorded from for at least 4 weeks. In order to compare two cohorts of 5 animals each (3xCR and GPC6 tumors), we had to combine animals with different calcium indicator types (3x thy1-GCaMP6s + 2x AAV-syn-GCaMP7f for 3xCR, and 3x CamKIIa-GCaMP8m + 2x AAV-syn-jrGeco1a for GPC6). We do agree that this could pose a challenge to correctly interpreting the results of figure 5B the way they were originally presented, since we were comparing animals with different promoters within the genotype groups. Therefore, we have removed this comparison. However, we believe that comparing across tumor genotype groups (but separately for slow and fast expansion states) is valid as both groups consist of the same mixture of pan-neural and pyramidal cell labelling. Figure 5B has been updated accordingly. We also removed the example cdf plots from 5A, as those are no longer relevant.

13. I can't make out the lower portion of the figure in Figures 5A and 5B or the axes in 5F and 5G. The resolution and magnification currently available do not permit reviewer assessment.

JM: We agree there were problems with figure resolution and clarity in the original, and now supply high-resolution figures in the resubmitted version.

14. The authors report that greater neuronal activity was observed during periods of fast tumor growth in both models – as evidenced by increased mean activity levels and greater amplitudes of the observed calcium events. Again, it is unclear if this is pan-neuronal activity or only excitatory CamKIIa neurons.

JM: see response to comment 12. We no longer make this claim.

Section: Tumor neuronal crosstalk depends on GPC6, tumor distance and growth rate.

15. This section is somewhat unclear – particularly the subsection which seeks to explain Figure 5F. This is compounded by the small size and poor resolution of the figures available to the reviewer.

JM: We agree that this subsection was not easy to follow, and the data presentation was suboptimal. In agreement with similar comments from other reviewers, we replaced this section of the figure with barplots & individual data points depicting the ratio between activity patterns near versus far from the tumor for the different activity metrics, both across tumor genotypes and fast versus slow tumor expansion periods. We think that this presentation provides a much easier and more intuitive way of deciphering precisely which experimental conditions led to significant differences in calcium activity as a function of tumor distance, genotype and expansion status.

Section: GPC6-dependent changes in cellular activity patterns inside versus outside the tumor margin.

16. As in earlier sections – it is unclear which AAV model was used, and thus it is unclear if the neuronal activity being tracked is pan-neuronal or specific to CamKIIa neurons only.

JM: Thank you for this observation. We have updated this figure, and listed all animals used here in the new supplementary table. We adjusted the description and discussion of these results to reflect the choice of measuring pyramidal cell activity only.

17. Again, the resolution and magnification of the images presented make it difficult to interpret 6A and the images in 6D.

JM: We agree and have generated images with improved resolution in the resubmitted version of the manuscript.

General:

18. Perhaps it is beyond the scope of this work, but it is not clear how you link the findings of decoupled glutamate and neuronal calcium levels to the results observed with 2P and mesoscopic imaging studies.

JM: We have added a schematic cartoon (suppl.fig 10) depicting the main conclusions of the study and how we think they might fit together. We are hopeful that this will add some clarity to the - admittedly complex - data we present.

Reviewer #3 (Remarks to the Author):

Review of Meyer et al – Nature Communications, 2023

In this study “Glioblastoma disrupts cortical network activity at multiple spatial and temporal scales”, by Meyer et al., the authors explore how neuronal activity and glutamate signaling is altered in 2 different models of GBM. GBM is a devastating disorder, has unique progressions based on tumor genetics, and desperately needs novel treatments. This study comes from an accomplished group with expertise in cancer, neuroscience, epilepsy, and in vivo imaging. The question is of high importance and studies are performed rigorously. The central findings presented are that the two GBM variants used (3xCR and GPC6) have different growth rates over time, invade different areas of cortex, and are correlated with unique changes in glutamate and neuronal activity. The authors highlight that there is an interesting and unexpected de-correlation between glutamate and Ca in the 3xCR mice, that there is a clear relationship between distance to the tumor and neuronal activity in the 3xCR mice but not the GPC6 mice, and that the timing of changes in neuronal activity are unique in each model. This is a very novel and cutting edge study and, as mentioned above, comes from experts in the field and is performed extremely rigorously. The central issue with the study, however, is the difficulty in understanding what imaging was done, how it was analyzed, and what it means. The author try to convey their complex findings to the reader but it is often extremely difficult to understand what the data is (missing figure legends, challenging text, no axes labeled) and how to interpret it (so many different assay, not sure what each means and if it is important). I applaud the authors on their models, questions, and imaging, but the presentation of the data needs improvement so that the reader can understand what they are reading. Overall, I remain highly enthusiastic for an exciting and rigorous study that if presented properly could be very impactful.

Major issues:

Q- In general, both the figures and the text are quite difficult to follow. The figures tend to lack labels, fonts are very small, and images are hard to see. The text is filled with jargon and the meaning of the data and analysis is often lost due to the dense and technical writing. The study’s impact would be significantly increased by improving the way the message is conveyed to the reader. In addition, the text changes between past and present tense which is difficult to read.

JM: We thank the reviewer for these observations, and have identified multiple sections in the text that needed improvement in terms of writing and clarity to convey the points we are making. We have adjusted all figures and improved image resolution and labeling, as we agree these were not optimized appropriately in the original version.

Q- A better explanation of what CV/day is and how it was used to analyze data in Fig. 3 is needed. It is not explained in such a way to favorably interpret the data. Why was CV/day used as a measure rather than total tumor area or something more simple? The issue that occurs with CV/day sit that when sequential timepoints are considered, if the tumor does not grow, the DCV/day value is negative (blue). This suggests the tumor is shrinking but I don’t believe that is the case. It seems like an overly complex and somewhat confusing way to quantify tumor growth.

JM: Thank you for this comment, we agree that this way of visualizing and quantifying tumor spread was not optimally explained in the manuscript. We deliberately chose to present the changes in tumor brightness in two distinct ways, as this was the only way to fully delineate how the tumors were changing over time. The changes in expansion rate we show in figure 3D were computed by calculating the change in tumor coverage of the cranial window as explained in the methods (we adjusted the language there to add clarification; see also similar questions by reviewers 1 and 4). This measure is likely closely related to tumor growth, but it is impossible for us to prove exactly how closely it follows overall growth (in 3 dimensions), so we adjusted the text throughout the manuscript to point out very clearly that this expansion rate only applies to the cortical surface where we imaged tumor and its interaction with the local milieu (see also our

reply to the very similar question by reviewer 4) The other measure, CV/day, is meant to reflect how fast the tumor is infiltrating the local TME at a certain time point. This measure would also capture pure tumor growth (i.e. proliferation) if one imagines the tumor to be a uniform sphere that steadily grows radially outward, but it is also very sensitive to small subpopulations of tumor cells moving through the parenchyma, which would be very hard to detect with other methods such as MRI, ultrasound or bioluminescence that could measure tumor growth in vivo over time. Since one of the motivating questions behind this study was to address the effects of GBM infiltration upon surrounding neural activity, CV/day was chosen as an appropriate measure of infiltration. We added clarifying language to the methods sections about this question, and specifically clarified that the blue color in fig 3A/B and 4A/B/C does not mean tumor shrinkage but merely movement (putatively away from where they were previously located in x- y- or z-direction).

Q- Perhaps a large section of the methods, where image analysis approaches are described should be moved to the results section. The analysis the authors use is custom made to deal with they type of data they have so it would be appropriate to describe their approaches in the results section.

JM: Thank you for this suggestion. We do agree completely that the data analysis methods are complex and require a great deal of explanation. However, we think that in terms of conveying the results and meaning of the observations in the clearest way possible, it is best to leave most of the detailed data analysis descriptions in the methods section. If we move significant parts of the methods to the results section, we fear that a lot of the data presentation (which this reviewer, in a previous comment, was correct to point out was overly dense and technical in some sections) would actually lose clarity and focus. Therefore, we choose to make significant improvements to the methods section in detailing the custom analysis procedures with great clarity, instead of moving it to the results (see also reviewer 4's question on 2-p data analysis, and the new supplementary figure 2). We did make some changes to the results section as well, improving clarity.

Q- The logic behind the uncorrelated glutamate and Ca signaling in the 3xCr mice is a bit hard to follow. Wouldn't Ca/neuronal activity still be increased if glutamate is increased? The author suggest that this reflect glutamate from a non-synaptic origin, which is very reasonable based on published GBM studies, but the increased glutamate should still drive local neuronal signaling. Also, Ca signals are likely dominated by somatic, not synaptic, Ca changes again underscoring the confusion in this interpretation.

JM: Thank you for this question, we agree that the apparent discrepancy between the glutamate and calcium dynamics in the 3xCR model is not easily explained and might appear counterintuitive. Of course, it has been conclusively shown in many different circumstances that an acute, local increase in extracellular glutamate leads to elevated neuronal spiking activity. However, there is also evidence that chronic inhibition of glutamate uptake can cause depolarization block, transmission failure, neurotoxicity, and degeneration (Rothstein et al, PNAS 1993; Pal, Cell Mol Life Sci 2018). Therefore, it is conceivable that the strong and sustained glutamate signal increase we see in 3xCR brains (which may be generated at least in part by dysfunctional glutamate reuptake of local astrocyte populations) is not mirrored by an equally strong increase in calcium signal, because ongoing degeneration counteracts the hyperactivity of neuronal populations. Besides peritumoral neuronal cell loss, we have also previously shown 3xCR-tumor induced global activation of microglia (Hatcher et al, JCI 2020). Microglia dysfunction could be another contributor to excess glutamate. Lastly, the calcium activity snapshots we show here depict brief baseline episodes of quiet wakefulness and do not contain information about fast activity patterns and metrics like the ones we analyzed in figure 5. Moreover, while a large portion of the fluorescence comes from neuronal cell bodies, these images also contain a lot of signal sources originating from dendrites and axons, so if there was a much stronger increase in synaptic activity, we hypothesize that it would be visible in these images. We have added caveats about the interpretation according to these thoughts to the discussion section.

Q- The interpretation of the Ca/glutamate relationship (uncorrelated in 3xCr mice) is also difficult to integrate with the fact that Ca activity is high near the tumor and decreases as you move away from the tumor border. Perhaps a model/cartoon would help the central points of the study be better conceptualized?

JM: As in the previous question, we agree that this result does not have an obvious explanation, but we stand by the data. One possibility is that the relatively high loss of GABAergic interneurons near the tumor margin elevates excitatory neuronal activity to such a degree that it dominates the effect of excess glutamate that is available near and far from the tumor.

Q- A model or cartoon depicting the overall changes across models, times, locations and assays (Ca/Glut/EEG) could be extremely useful. There are so many changes presented that it is challenging to synthesize it all. In addition, A small cartoon or laminar picture may help the reader better interpret the laminar data shown in Fig 3E.

JM: Thank you for this suggestion, a summary cartoon has been made and added as suppl. Fig 10.

Q- I have read and re-read Fig 4 and I honestly do not understand what is presented. The Glutamate and calcium image, I believe, are averages of periods of quiet restfulness (300 sec) and then DF/F is calculated and the CV is computed from within that data and plotted? Are the B&W fluorescent images DF/F ? Then below are the growth images of the tumor, but because they are color scaled and presented in an equally difficult way to follow (please see above) connecting what is shown in the top and bottom of each figure is extremely challenging. Finally, there are no labels on axes in Fig 4E and F again leading to the readers wondering what they are looking at. This is an extremely complex and high quality data set. I have no doubt of the rigor and precision that went into collecting this amazing data. It deserves a better presentation that allows the reader to understand what they are looking at.

JM: Thank you for this comment, we do apologize for the confusion about this figure and lack in clarity. We have modified the relevant methods sections to precisely clarify what it is we are showing in this figure. Briefly, figure 4 shows ONLY calcium and glutamate data (whereas figure 3 shows ONLY tumor data). The B/W panels in fig 4 A/B/C are snapshots generated by averaging 2 seconds of baseline calcium/glutamate fluorescence from 3 example animals over time (A: 3xCR calcium animal, B: 3xCR glutamate animal, C: GPC6 glutamate animal). We removed the GPC6 glutamate data, because we agreed that 2 animals were not sufficient to draw conclusions from, even though we were originally only showing no difference between early and later time periods in GPC6 glutamate (as opposed to 3xCR tumors). We chose to compare these baseline snapshots between calcium and glutamate signals, and not the $\Delta F/F$ converted calcium signals, because it would have been challenging to interpret glutamate dynamics after a similar $\Delta F/F$ conversion, since different biophysical processes underlie their rapid fluctuations. In other words, here we really focus on slow changes in glutamate accumulation in comparison with calcium baseline signaling, which in the mesoscale images reflects a mixture of somatic and synaptic spontaneous activity. Importantly, using different snapshots of calcium fluorescence from the same recordings did not change the outcome of our analysis.

Looking at the snapshots by themselves, however, would not give us much information about the dynamic changes in calcium and glutamate levels. Therefore, we computed the coefficient of variation between time points, and normalized by the number of days in between to yield the value of CV/d for each pixel. The colored panels underneath each row of B/W images show the result of this computation, and the numbers above each colored panel correspond to the time points used to compute them.

Q- Specific issues with Figure 4: How were Ca and glutamate images generated for Fig 4? Were static images used or were images generated based on some extrapolation from time series data? Static images are much less informative for Ca and glutamate since signals are so dynamic in time. The Y axes need to be labeled in Fig 4 E and F. In Fig 4 – only 2 glutamate imaging GPC6 mice were used? N = 2 is not sufficient for statistical analysis.

JM: Thank you for this question. We did use averages of only 2 sec of fluorescence for the calcium and glutamate image analysis in figure 4, which were recorded during periods of quiet wakefulness without active running or whisking, and no visual stimulation. The reason why we chose this approach was because the calcium and glutamate indicators have very different temporal dynamics, and therefore it would be hard to interpret a comparison of calcium $\Delta F/F$ versus glutamate $\Delta F/F$. Furthermore, we wanted to analyze slow changes over time and compare the very obvious and strong glutamate signal change over time with the baseline calcium signal which does not change as dramatically. We do think this comparison is meaningful even though it is just a “snapshot”, and it also shows a significant difference between 3xCR and GPC6 calcium in early versus late disease stages. In the subsequent sections of the paper we convert all calcium image sequences into $\Delta F/F$ traces and carefully analyze the “activity metrics” as defined in the methods, to analyze changes in fast activity patterns, which reflect changes at different time scales.

Q- How was the fast/slow growth cut off determined for Fig 4? Based on fig 3, it seems that there is no slow growth period for 3xCR as it grows consistently fast at all stages.

JM: We apologize for the lack of clarity here. We tried to improve the resolution of figure 3, which actually does show several 3xCR data points corresponding to slow expansion (close to the x-axis between P50 and P90), as well as just below the black line marking the division between slow and fast expansion between P90 and P130. We chose the expansion rate of $10e5 \text{ um}^2/\text{d}$ to split both the 3xCR and GPC6 data into relatively equal parts without over representing one tumor type or the other in either the slow or fast expansion group.

Q- In figure 5B, GPC6 tumors show no difference in events/sec in fast vs slow growth phases but in Fig 5C there appears to be as significant difference in the same measure? Please clarify. Is everything in Fig 5 Ca imaging? No glutamate?

JM: Thank you for pointing out this apparent contradiction. In Fig 5B we meant to show widefield analysis examples of a 3xCR and a GPC6 animal, and as an example of one of the activity metrics we compute, show the events/sec of all pixels in slow vs fast expansion recordings as a cumulative distribution function. This was supposed to show that we saw larger discrepancies in 3xCR than in GPC6 tumor animals. Fig 5C showed pooled data from several animals, all using calcium but no glutamate indicators. As described earlier, we realized that we unintentionally pooled data from animals with different types of calcium indicators without ensuring they were represented equally. We do not believe it is fair to compare slow with fast growth periods inside these mixed groups, therefore we have adjusted Fig 5B to only show comparisons of 3xCR against GPC6 groups, which contain equal numbers of animals labeled with pan-neuronal and pyramidal neuron-specific calcium reporters.

Q- How is seizure activity correlated with the different phases of tumor growth, glutamate signaling, and Ca activity? Is there a meaningful relationship between seizure progression and any of the assays? Is the progression of seizure or any seizure phenotype different between the 2 models?

JM: Thank you for this important question. We have previously shown (Yu et al, Nature 2020, fig 5e/f) seizure and EEG spike correlation with age for the 3xCR model (as well as with additional overexpression of GPC3, another member of the glypican family). Importantly, those previous experiments were carried out

using continuous EEG monitoring for several weeks per animal. This way, spontaneous seizures, which in GBM carrying mice occur rarely (typically only up to 1-3 times per day), could be reliably detected. In the present study, continuous EEG monitoring was not possible since the approved imaging protocol only allows for up to 6 hours of recording per day. Therefore, we do not include any quantitative seizure or spiking analysis in this study, as the data would have suffered from undersampling issues, making a comparison between models difficult. However, we did use identified seizure time points to exclude post-seizure activity from our analysis of spontaneous, quiet wakefulness activity, as seizures themselves acutely alter ongoing activity for substantial periods of time (see figure 1). In principle, given the fact that seizures are episodic events and often cluster, anything less than 24/7 monitoring can result in invalid comparisons.

Q- Why do the authors believe there is such a different relationship between tumor location and neuronal activity in the 2 models? That seems to be one of the central findings.

JM: We think that tumor genomics, rather than location (once the tumor invades the neocortex) is a key factor. While we do not claim to fully understand the mechanism, the evidence shows a distinct difference in the way our tumor models affect peritumoral neuronal activity depending on the addition of GPC6 expression. Specifically, it appears that the difference in activity near versus far from the tumor is more pronounced when GPC6 was expressed. At the same time, early mortality was higher in GPC6 expressing tumor mice, fast tumor expansion was seen earlier than in 3xCR tumor mice, and baseline calcium was elevated early. However, 3xCR tumors generally retained a high expansion and infiltration rate, and continued to affect TME activity over time, so we think that it is possible that 3xCR tumors, on average, remodel their environment more persistently to increase activity. We previously showed that 3xCR tumors cause some neuronal cell death around the tumor margin, preferentially of parvalbumin-expressing GABAergic interneurons (Hatcher et al, JCI 2020). Reduced inhibitory drive near the tumor could account for the relatively high activity we saw there. However, we do not know if GPC6 expression increases neuronal cell loss, and studying this in detail would be outside the scope of the present work. We added language reflecting these caveats to the discussion.

Minor Issues:

Q- Line 471-472 – “When we quantify all the measured events in a given frame during fast and slow growing stages in our 3xCr brain (Fig 5A), we observed 13% fewer events/sec.” In which group? Unclear

JM: This figure was updated, and the text was adjusted accordingly (see comment above). The cited lines are no longer in the manuscript.

Q- Was all imaging data taken from periods of animal stillness? How were the behavioral state quantifications integrated into the analysis?

JM: We have added some clarifications to the methods section to ensure our analysis of periods of quiet wakefulness is described appropriately. We recorded whisking (visually using an infrared camera), and locomotion (using a velocity encoder attached to the axle of the running wheel) throughout each recording. We identified periods of running and whisking and excluded those from any further cortical activity analysis.

Q- What does “Tumor neuronal crosstalk” mean? This is an example of jargony terms that should be avoided.

JM: In the relatively young field of “cancer neuroscience”, to our knowledge this term is used widely both in original papers and in review articles, meaning “feedback interactions between tumor cells and neurons”. We have defined this phrase in the text.

Reviewer #4 Overall

The work by Meyer et al. characterizing the influence of glioblastoma on neuronal changes tackles a set of very important and challenging questions in the field with a set of novel and exciting approaches. The authors focused on the spatial and temporal differences occurring in neurons between two mouse models of glioma, one with and one without an addition of GPC6. The authors found noteworthy results in synapse markers, widefield glutamate and calcium changes, and single neuron event rate changes. There are a number of claims and conclusions made in the paper that are not directly supported by the presented evidence, as detailed below. Additionally, there are a number of clarifications that need to be made regarding the data analysis and interpretations of results before publication. The work is exciting and novel and would be a great addition to the literature once properly revised.

Recommendation: Revise

Major Comments:

General:

- Across the entire paper the y-axes on the plots are not consistently labeled. This needs to be fixed

JM: Thank you for this observation. We have adjusted the y-axis labels across all figures for improved consistency.

- There seems to be a large difference between the time points along tumor progression that various analyses are done, which prevents one from understanding how the different analyses can be viewed together. For example, the staining performed in figure 2G-H looking at synaptic changes is done at P30 and the functional analyses are performed P40-P120. This needs to be addressed. If available, they should provide the histological analyses of the late stage tumors. At the very least they should discuss the limitations that this discrepancy puts on any correlation between the histological and functional analyses.

JM: Thank you for bringing this to our attention, we agree that there were some inconsistencies in the manuscript regarding the analysis of data taken at different time points and their interpretation, and we have clarified the related text passages throughout the manuscript. Unfortunately, we do not have histological data or saved tissue from the imaged brains at later time points, however it is also not clear that adding those data acquired from new animals would significantly improve our main conclusions. The synaptic staining was performed on purpose at P30 (much like in previous published papers by Yu et al (reference #17) and J Lin et al (reference #4), because at this early time point there is a higher chance of measuring the pure effect of GPC6 overexpression versus a combination of effects arising from interactions between the tumor and the microenvironment as well as secondary effects arising from different heterogeneous proliferation and invasion rates, and possible compensatory mechanisms occurring in the microenvironment such as peritumoral edema and cell death. We added statements to the text (underlined) clarifying our approach and explaining the limited applicability of the P30 staining data. to early time points only

- There is a presumed starting point of this paper that there is a known increase of synapses at the margin of glioma, a “hypersynaptic network”. This evidence is not shown in this paper and I know of no convincing evidence in the literature of direct measurements of

increased synaptic density at the glioma margin. To provide direct support for this claim, they should provide comparative analysis to control brains.

JM: We agree that this was not stated precisely enough, so we added clarifying language here to reflect more accurately the data we have published previously in this exact same IUE glioma model (J Lin et al, Yu et al, Hatcher et al). Instead of “hypersynaptic”, which could be interpreted as “more synapses per volume”, we describe the TME as having a “remodeled” distribution of synapses, both in density and more crucially in type. We cite additional evidence for this from other studies that analyzed human as well as murine TME tissues.

- Methods:

The authors need to provide more details on how they performed their 4 “metrics” of analysis for the 2P data set.

- Specifically, this figure should address the following issues which are currently unclear:
 1. What is meant by “overall (summed) $d\Delta F/F$ ”? What constitutes an “event” and what does this “event” look like?

JM: The output of the deconvolution algorithm we applied here (adapted from Suite2p-code openly available on Github) does not precisely “binarize” the signal into exact, instantaneous time points of “spike” or “no spike” as one would obtain from single-cell patch-clamp recordings. Rather, it is an estimate of the current probability that one or multiple spikes were produced within a time bin corresponding to the inverse of the scan rate (typically ~10-12 Hz for spiral scan, and 15-30Hz for resonant scan). Therefore, bursts of spikes occurring within ~33-100ms generally show up as one distinct “event”, as do single spikes. As far as we know, there is no strict consensus as to the general meaning of a spike burst, its duration, intra-burst frequency, etc. The most meaningful analysis we can perform is to compare the measured ranges of these parameters to each other and to known baseline values. We added a supplementary figure visualizing this process (see below).

2. How did the authors determine the “duration” of a calcium transient on deconvolved data? Presumably when deconvolved, the calcium event will be discretized into a discrete number of single spike times.

JM: See explanation above. We added a supplementary figure (#2) depicting raw calcium versus $\Delta F/F$ versus deconvolved event shapes in a subset of cells from a GPC6 animal to visualize how we measured metrics such as event rate, amplitude, etc. This should clarify that event duration is an estimate of spike/ burst length, with an approximate minimum intra-burst interval of ~100ms, equivalent to a minimum intra-burst firing rate of ~10Hz. Of course this parameter could have been chosen differently, but based on previous studies we believe that this is a physiologically reasonable time scale to choose. We actually do not show event duration from 2-photon imaging datasets in the manuscript because it did not provide meaningful information about the activity patterns we observed, as opposed to event amplitude which tells us about instantaneous spike rates produced by different neuronal populations.

3. How did the authors account for the slow decay of calcium fluorescence in determining “coactive clusters” ie how was the temporal window determined under which clusters were coactive? Neither of the cited works for the clustering coefficient analysis include an application to calcium imaging data

JM: We used the deconvolved data with an estimated resolution of 33-100msec, and applied time binning of 100 msec when applicable. To determine correlation coefficients, which were the basis for cluster detection, we followed previously published approaches (e.g. Meyer et al 2018). We added 3 references of studies that used two-photon calcium imaging and cluster/ensemble detection algorithms based on the same papers we based our approach on (Rubinov et al. 2010; Rubinov et al. 2011)

Figure 2:

-
For the data in Fig2G-H, it would be very informative to also plot PSD95, VGlut1, Gephyrin, and VGAT levels alone in these different

JM: Thank you for this suggestion. Presumably the reviewers mean “in these different groups”. We have added greyscale panels for Psd95, Vglut1, Gephyrin and Vgat respectively in fig 2 g-h.

Figure 3:

- It appears that the fluorescence tumor signal disappears after P83 in Fig 3B, which could be due to numerous experimental reasons other than tumor growth changes. The conclusion that the data in figure 3 “reveal that 3xCR tumors relentlessly infiltrate cortical territory throughout disease progression, whereas GPC6 tumors first infiltrate aggressively, but grow less dramatically later once they reach cortical layers 1-4” needs to be corroborated by tissue-based analysis of the brains.

JM: We agree that based on our observations, these conclusions were not presented and stated in a convincing way. We no longer make any claims about changes in overall tumor growth or size, since to be precise, it is not overall growth but expansion of the tumor cell fields within the imaging window that we are analyzing. In our opinion, this does not diminish the overall main messages of the study about the interaction of tumor cells and cortical neurons. Therefore, we adjusted the text accordingly and replaced the section in parentheses with “indicate that 3xCR tumors reach more rapid infiltration rates than GPC6 tumors locally at the cortical surface. GPC6 tumors appear to infiltrate aggressively at first, but less dramatically later once they reach cortical layers 1-4.” We also adjusted the corresponding sections in the discussion and other results sections to match the change in focus from “growth” to “local” infiltration or expansion patterns. When describing the “growth rate” visualized in Fig 3, we changed this to “expansion rate” to be more precise since proliferation and migration are separate processes. Again, we adjusted any corresponding section in the text to reflect that we are only making claims about local cortical expansion, not the entire tumor. We do remain convinced that these comparisons are meaningful, because based on previously published data on the 3xCR tumor model it is highly unlikely that only local cortical infiltration and expansion rates would be different between 3xCR and GPC6 tumors, and in all other parts of the tumor this would not be the case. Even if that were true, though, we are not generalizing our findings to other parts of the tumor in the paper.

- Given the concerning reduction in fluorescence signal across all measurements – tumor fluorescence, calcium, glutamate – in the GPC6 model in the later time points (after ~P60) – it would be very important for the authors to plot all the signals for each animal together and see

if they reduce together as time progresses. Perhaps the changes being seen are due to the quality of the cranial window decreasing. This concern could be addressed by correlating the last time-point of in vivo imaging of tumor burden with the postmortem histological analysis of tumor size.

JM: We have images, both from the simultaneously acquired calcium indicator fluorescence in the widefield mode, and the cellular-resolution two-photon images, which demonstrate that this reduction in fluorescence does not stem from an issue with the optical clarity of the window (which we take enormous pains to preserve). We added a supplementary figure (#6) showing all time points imaged in this animal for both channels – tumor and calcium indicator - to visualize this. Note that we only had tumor and calcium OR tumor and glutamate indicators in each animal, not all three. We checked all 3xCR and GPC6 images again and did find one other GPC6 animal that showed a similarly strong drop off in fluorescence amplitude at a late time point. Just to err on the side of caution, we re-processed all images from these two animals by first measuring the tumor/calcium independent background signal from inside a large pial vein, then normalizing each image by its background signal. Figure 3 was adjusted accordingly. In addition, we removed the last 3 time points from the 3xCR and GPC6 examples in A and B. With respect to tumor infiltration (statistics in C) we are not claiming any specific differences between early and late stages, we are only saying that GPC6 tumors generally did not reach infiltration speeds that we saw in 3xCR animals. Moreover, to be more precise, we are not claiming changes in tumor growth rate, as it is true that the tumor could be growing at different speeds elsewhere in the brain where we cannot observe it directly through the cranial window. In previous serial studies (Hatcher et al, JCI 2020), fluorescent images of the tumor reveal a markedly uneven pattern of bulk tumor growth). Even though we believe that the size of the tumor at the surface correlates with overall size, we are now more accurately describing the measurements as tumor coverage, since this better reflects that we are really only measuring local growth at the cortical surface.

Figure 4:

- The presentation and explanations of the data in figure 4 is confusing and difficult to interpret. The data clearly demonstrates dynamic changes in glutamate and calcium but fails to demonstrate any clear patterns or correlations between the two. Interpretations beyond that seem speculative.

JM: We agree and have clarified our observations in the text. The main result of this analysis is that we observed a clear disconnect between the calcium and glutamate dynamics, as well as differences in calcium dynamics over time versus no significant changes over time for glutamate (quantified in figure 4D/E, albeit this is an indirect correlation since, as mentioned above, we never expressed calcium and glutamate indicators in the same animals). We clarify this in the results and methods sections. We adjusted the text in the results section to clarify that we are not making any direct correlations with tumor growth in this analysis, as we are not showing tumor images in figure 4.

- The histologic analysis performed in Figure 2 is done much earlier than the analyses performed in figures 3 and 4, and no quantitative analyses was done to correlate the two. Any conclusions drawn by connecting these two analyses are speculative and should be framed as such. For example, manuscript lines 459-460.

JM: We agree that the results from figure 2 do not directly map onto the results in figures 3 and 4. We

removed the text in lines 459-460. To emphasize the speculative nature of this evidence, we adjusted the corresponding text in the discussion (originally lines 656-658) to match the results.

Figure 5:

- The results of figure 5D-G are confusing and need to be clarified more carefully in both the manuscript and figure. For example, in this statement – “In both 3xCr and GPC6 tumor brains, there was a significantly higher event rate proximal to the tumor (p-values .02 and 2e-6, respectively)” – what is the meaning of this comparison to this null distribution? Why not directly compare event rates near and far from the tumor between the two models?

JM: We agree that the presentation of the data in figure 5D-G needed to be improved by showing the comparison the reviewer suggested. We computed this comparison to update the figure and the corresponding text as well as a description of this new comparison in the methods and moved the original figure panels as additional results to supplementary figure 8.

Figure 6:

- There are 4 metrics for each subplot ($\Delta F/F$, events/s, amplitude, clustering) and 3 subplots for a total of 12 comparisons. However, there is only data presented for 4 out of these 12 comparisons. The authors should explain why these other comparisons were not considered.

JM: We agree that these results needed to be presented more clearly. For figure 6b) we now display comparisons between groups of neurons inside the tumor margin (0-1mm from the edge of the solid main tumor mass) and outside the margin (1-2mm from the edge of the solid main tumor mass), separately for 3xCR and GPC6 tumor animals. For 6c) we compare recordings from 3xCR against GPC6 animals inside the tumor margin. In order to avoid displaying unnecessary information, the grey bar plots in the original manuscript only showed the effect sizes (percent change between inside vs outside the margin, and 3xCR vs GPC6 inside the margin, respectively) for comparisons with statistically significant differences. Instead of barplots with error bars, we showed the actual data distributions as violin plots for those significant comparisons. In order to be more transparent, and to align the visual presentation of the data with figures 5 and 7, we now show bar plots with individual data points instead of the gray bars, and indicate which comparisons were statistically significant. We also indicate that only meaningful comparisons (inside versus outside for each genotype, and 3xCR vs GPC6 for inside and outside the margin, respectively) are highlighted.

Figure 7:

- The way the data is organized in figure 7 does not allow the reader to directly compare how each metric ($\Delta F/F$, events/s, amplitude, clustering) changes in the animals across time (early vs. middle vs. late). It would be much easier to understand as each metric simply plotted through time (early, middle, late) with clear indications of which time points have statistical differences. Currently, there are no statistics indicated on the figure.

JM: As in figure 6, we replaced the gray barplots showing only significant comparisons with bars + individual data points across time points and genotypes. We clarified this in the main text and methods.

Minor Comments

- From the low magnification images in Fig 2G-H, it is difficult to orient oneself to where in the brain the peritumoral analysis has been done. Authors should please provide more clear low mag images to orient where in the brain the tumor and imaging fields are, so that one can properly assess whether the authors have taken brain regions differences into account in this analysis. Additionally, in this figure, the ordering of the groups in the boxplots should match the ordering of the columns of the representative images.

KY: We thank the reviewers for these suggestions. In these experiments we assess the effect of tumors and the synaptic microenvironment. While our comparisons were the effects of GPC6 tumors, we did not take into account that the synaptic microenvironment may also differ across normal cortical brain regions. Unfortunately, the specific brains used for these experiments have aged beyond their ability to accurately address these concerns. Since the investigator who performed these specific experiments has since moved on and what data that was left behind would not answer this, to directly address these concerns would require repeating the entirety of the experiments, from generating tumor bearing mice, aging these mice, and processing the resulting brains. As alternative to performing these experiments, we offer the reviewer the following assurances: 1) While we did not explicitly state this in the manuscript, we routinely perform these analyses in cortical tissue. We have amended the manuscript to explicitly communicate that these analyses were performed in various cortical regions. 2) We have performed similar studies across multiple published manuscripts (Yu et al, Nature 2020, PMID: 31996845; Yu et al, Neuro-Oncol 2023, PMID: 36044040; Curry et al, Neuron 2023, PMID: 36787748; and Huang-Hobbs et al, Nature 2023, PMID: 37380778) and observed consistency in our approach. Regardless of which cortical region we analyze (provided it is within the tumor margin as depicted in our low magnification images), we consistently and reproducibly observe these patterned changes. We include this in the text.

- Authors need to provide more details about how the immunofluorescence images in Fig 2G-H were acquired and analyzed for colocalization.

KY: We added the following text to the methods:

Quantification of peritumoral synapses

Excitatory and inhibitory synapses at the tumor margins were quantified as previously published (Yu et al, Nature 2020, PMID: 31996845; Yu et al, Neuro-Oncol 2023, PMID: 36044040; Curry et al, Neuron 2023, PMID: 36787748; and Huang-Hobbs et al, Nature 2023, PMID: 37380778). Briefly, frozen brains were sectioned to 40 μm thickness and stained with the aforementioned pre- and post-synaptic markers for excitatory (VGlut1/PSD95) and inhibitory (Vgat/Gephyrin) synapses. Confocal images were taken with a Zeiss LSM 880, and functional synapses were identified by the colocalization of the pre-/post- pairs using the SynapseCounter plugin in ImageJ (Dzyubenko et al, J Neurosci Methods 2016, PMID: 27615741). We used a control section stained only with secondary antibodies to establish threshold levels for discerning positive staining compared to no background fluorescence.

- The use of “early,” “middle,” “late” timepoint labels is very confusing and inconsistent throughout the paper. On line 352 of the manuscript authors say “(early (P40-49), mid (P-50-59)

and late (P60-100), but later on lines 437 the early is defined as P56 – P69. What time point would the analyses in figure 2 at P30 be considered?

JM: Thank you for this comment, which is related to other concerns mentioned above. We added extensive clarification for this in the results and methods sections.

- What do the red and orange colors indicate on the raster plots in Figure 6A and 7A? If amplitude of signal, include color scale info.

JM: Thank you for catching this, these colors correspond to deconvolved calcium event amplitudes, and we added color scales to figures 6 and 7.

REVIEWERS' COMMENTS

Reviewer #1 (Remarks to the Author):

The authors have successfully addressed all of my inquiries in a satisfactory manner. I am happy to recommend publication of this work.

Reviewer #2 (Remarks to the Author):

The revised version of the manuscript is an improved product. I still have several reservations about the study and while beyond the scope of the investigation a mechanistic link explaining how GPC6 expression is directly or indirectly influencing these longitudinal non-linear peritumoral neuronal alterations is not offered. However, I acknowledge that they have addressed – either in part or in full - my main concerns regarding:

- 1) Evidence of GPC6 expression in tumor cells.
- 2) Layer specificity of the subcortical infiltration pattern.
- 3) Neuronal cell-type specificity of the calcium signal under investigation in the growth rates experiments.

Despite that, I do think that several open questions remain (as is the case with all manuscripts), but most notably the relevance to human tumors, as IVY-GAP data suggests GPC6 expression is confined to the tumor core, not the periphery where tumor cells and neurons intermingle.

Reviewer #3 (Remarks to the Author):

The authors have significantly updated the analysis and text of this study. I already felt this was a rigorous and comprehensive study and most of my previous issues were caused by the difficulty of

communicating and presenting such a complex study. The authors have gone to great lengths to improve this aspect of the study and have addresssed all my concerns. Congrats on an excellent study.

REVIEWERS' COMMENTS

Reviewer #1 (Remarks to the Author):

The authors have successfully addressed all of my inquiries in a satisfactory manner. I am happy to recommend publication of this work.

Reviewer #2 (Remarks to the Author):

The revised version of the manuscript is an improved product. I still have several reservations about the study and while beyond the scope of the investigation a mechanistic link explaining how GPC6 expression is directly or indirectly influencing these longitudinal non-linear peritumoral neuronal alterations is not offered. However, I acknowledge that they have addressed – either in part or in full - my main concerns regarding:

- 1) Evidence of GPC6 expression in tumor cells.
- 2) Layer specificity of the subcortical infiltration pattern.
- 3) Neuronal cell-type specificity of the calcium signal under investigation in the growth rates experiments.

Despite that, I do think that several open questions remain (as is the case with all manuscripts), but most notably the relevance to human tumors, as IVY-GAP data suggests GPC6 expression is confined to the tumor core, not the periphery where tumor cells and neurons intermingle.

Reply: Thank you for the thorough examination of our study, and for emphasizing this potential difference between our model and the human data sources we used. While we agree that a preferential expression density near the tumor core in humans is a valid potential caveat for the interpretation of our results, there are also examples in IVY-GAP that do show lower but still significant GPC6 expression in infiltrating tumor zones. Nonetheless, to be conservative and not overstate our findings, and in light of the fact that in the GEPIA database there is no information on the exact location for each tumor sample underlying the GPC6 transcript and survival data we show, we added the following statement to the text:

“Of note, in our model GPC6 expression was introduced into all tumor cells, whereas in the human samples GPC6 was likely not equally as homogeneously expressed throughout the tumor.”

Reviewer #3 (Remarks to the Author):

The authors have significantly updated the analysis and text of this study. I already felt this was a rigorous and comprehensive study and most of my previous issues were caused by the difficulty of communicating and presenting such a complex study. The authors have gone to great lengths to improve this aspect of the study and have addressed all my concerns. Congrats on an excellent study.